# LRRK2 kinase regulates α-synuclein propagation via RAB35 phosphorylation

Eun-Jin Bae[1], Dong-Kyu Kim[1], Changyoun Kim[2,3], Michael Mante[3], Anthony Adame[3], Edward Rockenstein[3], Ayse Ulusoy[4], Michael Klinkenberg [4], Ga Ram Jeong[5], Jae Ryul Bae[5], Cheolsoon Lee[6], He-Jin Lee[6], Byung-Dae Lee[7], Donato A. Di Monte[4], Eliezer Masliah[2,3,8] & Seung-Jae Lee[1]

Propagation of α-synuclein aggregates has been suggested as a contributing factor in Parkinson's disease (PD) progression. However, the molecular mechanisms underlying α-synuclein aggregation are not fully understood. Here, we demonstrate in cell culture, nematode, and rodent models of PD that leucine-rich repeat kinase 2 (LRRK2), a PD-linked kinase, modulates α-synuclein propagation in a kinase activity-dependent manner. The PD-linked G2019S mutation in LRRK2, which increases kinase activity, enhances propagation efficiency. Furthermore, we show that the role of LRRK2 in α-synuclein propagation is mediated by RAB35 phosphorylation. Constitutive activation of RAB35 overrides the reduced α-synuclein propagation phenotype in *lrk-1* mutant *C. elegans*. Finally, in a mouse model of synucleinopathy, administration of an LRRK2 kinase inhibitor reduced α-synuclein aggregation via enhanced interaction of α-synuclein with the lysosomal degradation pathway. These results suggest that LRRK2-mediated RAB35 phosphorylation is a potential therapeutic target for modifying disease progression.

[1] Departments of Biomedical Sciences and Medicine, Neuroscience Research Institute, Seoul National University College of Medicine, Seoul 03080, Korea. [2] Molecular Neuropathology Section, Laboratory of Neurogenetics, National Institute on Aging, National Institutes of Health, Bethesda, MD 20892, USA. [3] Department Neurosciences, School of Medicine, University of California, San Diego, La Jolla, CA 92093, USA. [4] German Center for Neurodegenerative Diseases (DZNE), Sigmund-Freud-Strasse 27, 53127 Bonn, Germany. [5] Department of Neuroscience, Graduate School, Kyung Hee University, Seoul 02447, Korea. [6] Department of Anatomy, School of Medicine, Konkuk University, Seoul 05029, Korea. [7] Department of Physiology, School of Medicine, Kyung Hee University, Seoul 02447, Korea. [8] Department of Pathology, School of Medicine, University of California, San Diego, La Jolla, CA 92093, USA. These authors contributed equally: Eun-Jin Bae, Dong-Kyu Kim. Correspondence and requests for materials should be addressed to S.-J.L. (email: sjlee66@snu.ac.kr)

Parkinson's disease (PD) is a common neurodegenerative disorder characterized by movement symptoms, as well as various non-motor symptoms[1]. PD belongs to a group of diseases known as synucleinopathies because its pathological characteristics include abnormal deposition of α-synuclein aggregates in the forms of Lewy bodies (LBs) and Lewy neurites (LNs) in surviving neurons[2]. The gene for α-synuclein, SNCA, has not only been linked to familial PD but is also strongly associated with sporadic PD[2]. Importantly, α-synuclein aggregates sequentially spread from a few discrete regions to wider regions in the brain as the disease progresses[3], and the pattern of aggregate spreading is somewhat correlated with clinical progression, increasing the possibility that the former is the main driver of disease progression.

A growing body of evidence suggests that cell-to-cell propagation of α-synuclein aggregates through interconnected brain regions is the principle mechanism underlying pathological aggregate spreading[4]. However, postmortem analysis of human brains and connectomics studies challenge the prion hypothesis of the aggregate propagation in PD[5]. Nevertheless, aggregate propagation is a real phenomenon that has been confirmed in numerous cell and animal models. Thus, studies of this phenomenon will provide insights into the etiopathology of PD. The aggregate propagation seems to be mediated by exocytosis of aggregates from donor neurons and subsequent aggregate endocytosis by recipient neurons[6], although a recent study suggested that tunneling nanotubes can enable the neurons to directly transport α-synuclein aggregates[7]. The internalized α-synuclein aggregates are trafficked through the endolysosomal pathway[8]. The current understanding of the propagation details and identification of key regulators of the process remains very primitive.

Mutations in leucine-rich repeat kinase 2 (LRRK2) are considered the most common cause of both familial and sporadic PD[9–12]. LRRK2 is a member of a multidomain protein family, which also includes leucine-rich repeat kinase 1 (LRRK1) (Supplementary Fig. 1). Domain structures of LRRK proteins are conserved between the isotypes and among different species. C. elegans has one LRRK gene, lrk-1, which has domain structures homologous to those of mammalian LRRK genes[13]. Overall structural conservation suggests functional similarity between LRK-1 and mammalian LRRKs. It is worth taking into consideration a recent report showing that two LRRK2 kinase inhibitors did not inhibit LRK-1[14]. However, the evidence provided in this study was indirect, and the direct effects of LRRK2 kinase inhibitors on LRK-1 remain to be determined. The most common mutation in LRRK2 is the G2019S substitution in the kinase domain, which leads to an increase in the kinase activity[15]. Interestingly, many PD patients with LRRK2 (G2019S) mutations exhibit α-synuclein-positive LBs[9,16,17], even though nearly half the LRRK2 (G2019S)-PD cases are LB-negative[18,19]. This suggests that LRRK2 mutations could be implicated in α-synuclein-induced neurodegeneration in PD. Overexpression of LRRK2 has been shown to exacerbate while deletion of the Lrrk2 gene suppressed synucleinopathy lesions and α-synuclein-induced neurodegeneration in rodent models of synucleinopathy[20,21]. Inclusion body formation that was triggered by administration of α-synuclein fibrils in neuronal culture and rat brain was also increased by transgenic expression of LRRK2[22]. RNAi-mediated silencing of lrk-1 expression led to neuron-to-neuron propagation of α-synuclein in C. elegans[23]. However, the mechanism by which LRRK2 exerts its action in synucleinopathy is not fully understood.

Here, we investigate the roles of LRRK2 and LRK-1 in the propagation of α-synuclein in cell culture, nematode, and rodent models. The current study demonstrates that LRRK2 can regulate α-synuclein propagation in a kinase activity-dependent manner

via phosphorylation of RAB35. Furthermore, our findings suggest that LRRK2 kinase activity can be targeted for the purpose of reducing synucleinopathy lesions.

## Results

**lrk-1 and Lrrk2 deficiencies reduced α-synuclein propagation.** We employed two independent in vivo models, C. elegans and rat models, which were deficient in lrk-1 and Lrrk2 genes, respectively. First, we used the C. elegans model, in which the N-terminal and C-terminal halves of Venus fluorescent protein were fused to α-synuclein protein (V1S and SV2, respectively) and expressed in pharyngeal muscle and the connected neurons, respectively (Fig. 1a). This model allowed for quantitative analysis of intercellular α-synuclein propagation between pharyngeal muscles and neurons through bimolecular fluorescence complementation (BiFC)[8]. We generated three independent BiFC transgenic lines each in wild-type (WT) (N2) and lrk-1 mutant C. elegans, with the levels of α-synuclein expression matched (Supplementary Fig. 2a, b). The Venus fluorescence was increased with aging in V1S+SV2 transgenic worms in the WT background (Supplementary Fig. 2c, d). However, in lrk-1 background worms, the fluorescence was not increased with aging, showing significantly reduced Venus fluorescence at day 13 compared with the WT worms (Fig. 1b, c and Supplementary Fig. 2c, d). Likewise, the number of Venus-positive inclusion bodies was greatly reduced in lrk-1(tm1898) (Fig. 1d and Supplementary Fig. 2e, f). We then examined various degenerative phenotypes in the transgenic lines. Consistent with a previous report[8], the number of both axonal blebs (Fig. 1e) and fragmentations (Fig. 1f and Supplementary Fig. 2g) was increased in N2; V1S+SV2 lines at old ages. This axonal damage was ameliorated by lrk-1 gene mutation (Fig. 1e, f). The age-dependent α-synuclein propagation was accompanied by behavioral deficits, in this case, a decline in the pharyngeal pumping rates (Fig. 1g). The lrk-1 mutation completely eliminated this behavioral phenotype (Fig. 1g). Finally, the decrease in life span in the BiFC transgenic worms was almost completely rescued by lrk-1 mutation (Fig. 1h and Supplementary Fig. 2h, i). Therefore, α-synuclein propagation and the associated degenerative phenotypes require the function of the lrk-1 gene in C. elegans.

We next tested the role of LRRK2 in a rat model in which neuron-to-neuron α-synuclein transfer and long-distance protein spreading are triggered by injection of recombinant adeno-associated viral vectors (AAVs) encoding human α-synuclein into the vagus nerve. In this model, targeted overexpression of the exogenous protein in the dorsal medulla oblongata triggers progressive caudo-rostral spreading of the protein towards the pons, midbrain and forebrain, affecting regions that are anatomically connected to the dorsal motor nucleus of the vagus (DMnX) and the nucleus of the tractus solitarius[24]. Experiments were carried out using Charles River Long-Evans wild-type (WT) controls and homozygous LRRK2-deficient (-/-) rats on the Charles River Long-Evans background strain[21]. The absence of LRRK2 in the latter group of animals was confirmed by RT-PCR and immunohistochemistry (Supplementary Fig. 3a, b). Rats were killed at 8 and 12 weeks after intravagal injection of AAVs. Neurotoxicity was assessed by stereological counting of DMnX neurons, and transduction efficiency was estimated by calculating the percentage of DMnX neurons overexpressing human α-synuclein[25]. Animals in the different experimental groups were matched based on lack of neurodegeneration and similar counts of transduced DMnX neurons (approximately 20% of the total neuronal number; Supplementary Fig. 3c, d). Axons immunoreactive for human α-synuclein were observed in brain regions rostral to the medulla oblongata of either WT or Lrrk2$^{-/-}$ rats,

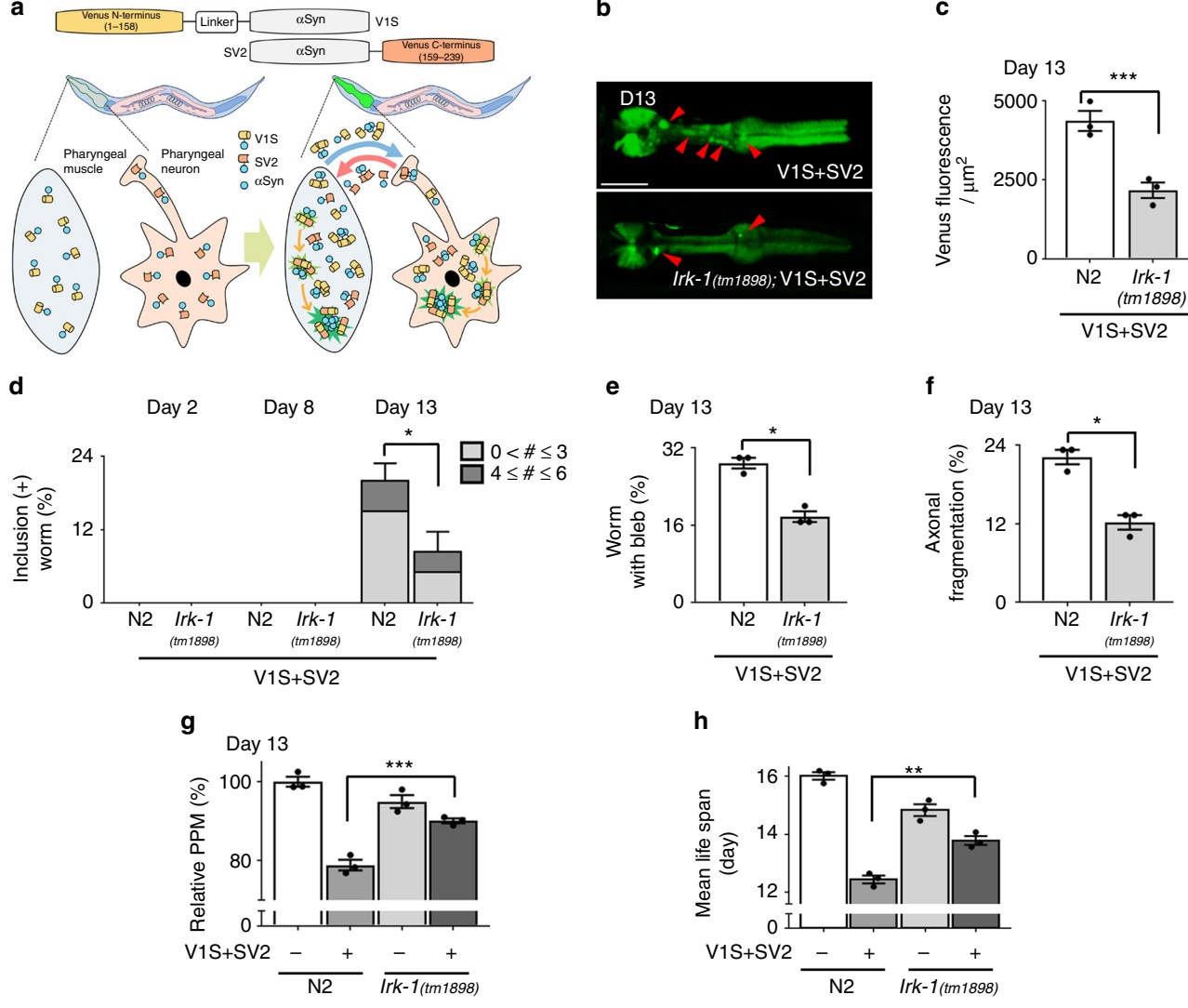

**Fig. 1** Effects of *lrk-1* deficiency in α-synuclein propagation. **a** Scheme of the propagation of α-synuclein propagation model in *C. elegans*. **b** Venus BiFC fluorescence in wild-type and *lrk-1(tm1898)* mutant worms at day 13. The red arrowheads: inclusions in pharynx. **c** Quantification of Venus BiFC fluorescence at day 13. Twenty worms for each line were used, *N* = 3, *** *P* < 0.001. **d** Venus BiFC-positive inclusions with aging. The color bars in graph (**d**) represent quantification of number of Venus BiFC-positive inclusions in each BiFC transgenic line. Twenty worms for each line were used, *N* = 3, * *P* < 0.05. **e**, **f** Nerve degeneration. Percentage of worms with axonal bleb (**e**) and with fragmented axonal processes (**f**) in URA motor neurons in each transgenic line at day 13. Thirty worms for each line were used (*N* = 3 in each group), * *p* < 0.05. Error bars represent SEM and *P* values, including * *p* < 0.05, *** *p* < 0.001, were calculated by unpaired, two-tailed Student's *t* test. **g** Relative pharyngeal pumping rates at day 13. Thirty worms for each line were used (*N* = 3 in each group). F(3, 270) = 65.620, *** *p* < 0.001 by one-way ANOVA with Tukey's post hoc test. **h** Life span analyses. One hundred worms for each line were used (*N* = 3 in each group). F(3, 9) = 91.860, ** *p* < 0.01 by one-way ANOVA with Tukey's post hoc test. All values shown in the figure represent mean ± SEM

indicating inter-neuronal spreading of the exogenous protein. Positive fibers were characterized by robust immunoreactivity within irregularly spaced varicosities (Fig. 2a). The number of labeled axons was determined and compared in the left (AAV-injected side) pons of WT *vs. Lrrk2^{−/−}* animals. In agreement with previously reported findings[24], a temporal progression of human α-synuclein spreading was indicated by higher counts at 12 weeks compared with the earlier time point (Fig. 2b). When the number of immunoreactive fibers was compared between WT and *Lrrk2^{−/−}* rats, decreased counts were observed in the latter group at both 8 and 12 weeks (Fig. 2b). To enhance the specificity and sensitivity of this comparison, unbiased measurements of length and density of human α-synuclein-positive fibers were taken in a pontine region encompassing the coeruleus/subcoeruleus complex and the parabrachial nuclei using the Space Balls

stereological tool[25]. Data at 8 and 12 weeks were pooled and analyzed together. The results showed that the mean total length and mean density of labeled axons were both significantly decreased in *Lrrk2^{−/−}* animals (Fig. 2c, d). A final set of analyses involved counting of immunoreactive axons in more rostral brain regions. Similar to the results in the pons, counts were found to be markedly lower in sections from the caudal (−69%) and rostral (−69%) midbrain and from the forebrain (−55%) of *Lrrk2^{−/−}* rats (Fig. 2e). Taken together, the rat data demonstrated that LRRK2 plays a critical role in cell-to-cell propagation and long-distance spreading of α-synuclein in vivo.

**LRRK2 kinase promotes α-synuclein propagation.** We then generated recombinant adenoviral vectors containing *LRRK2* WT

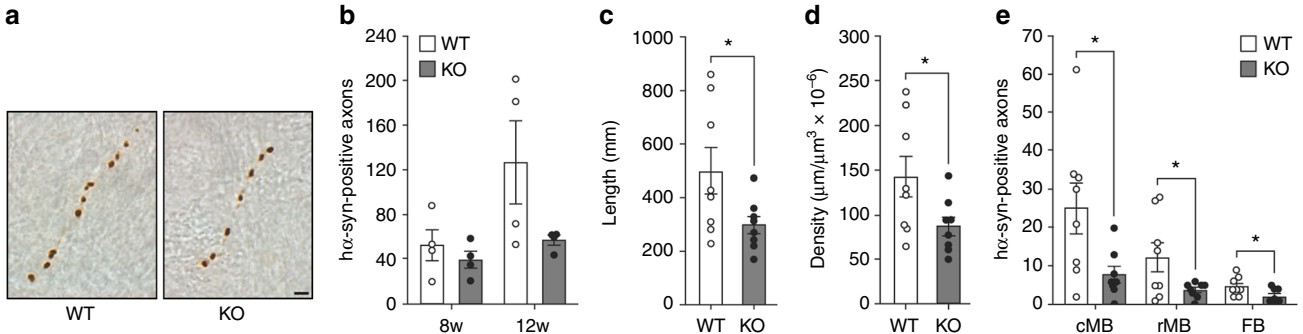

**Fig. 2** Effects of *Lrrk2* ablation in α-synuclein spreading in rats. Spreading of α-synuclein is decreased in LRRK2-deficient rats. Wild-type (WT) and LRRK2-deficient (KO) rats received a single injection of AAVs encoding human α-synuclein (hα-syn) into the left vagus nerve. Spreading of α-synuclein was analyzed 8 and 12 weeks later. **a** Representative images of axons stained with an antibody against human α-synuclein in the left (AAV-injected side) pons of a WT and a KO rat killed at 12 weeks post-AAV injection. Scale bar: 5 μm. **b** The number of axons immunoreactive for human α-synuclein was quantified in the left pons (Bregma: −9.48 mm) of WT ($N = 4$/time point) and KO ($N = 4$/time point) rats. Error bars represent SEM. **c, d** Length (**c**) and density (**d**) of axons immunoreactive for human α-synuclein were estimated in the left pons in an area encompassing the coeruleus/subcoeruleus complex and the parabrachial nucleus. Samples were obtained from rats killed at 8 ($N = 4$ WT and $N = 4$ KO) and 12 ($N = 4$ WT and $N = 4$ KO) weeks post-AAV injection and, for each animal, measurements were carried out on three separate pontine sections. Data at the two time points were pooled and analyzed together. Error bars represent SEM, * $p < 0.05$ by unpaired, two-tailed Student's *t* test. **e** The number of axons positive for human α-synuclein was quantified in the left caudal midbrain (cMB; Bregma: −7.8 mm), rostral midbrain (rMB; Bregma: −6.0 mm) and forebrain (FB; Bregma: −2.4 mm) of WT ($N = 4$ at 8 weeks and $N = 4$ at 12 weeks) and KO ($N = 4$ at 8 weeks and $N = 4$ at 12 weeks) rats. Error bars represent SEM, * $p < 0.05$ by unpaired, two-tailed Student's *t* test

or the G2019S or D1994A mutant. We confirmed the ectopic expression of LRRK2 WT and its mutant variants in differentiated SH-SY5Y neuroblastoma cells (Supplementary Fig. 4). The overall expression levels were similar among the LRRK2 variants. The intracellular distribution of LRRK2 proteins was indistinguishable among the experimental groups. Note that most of the SV2 cells were infected with the virus and represented approximately 50% of the total cells in the co-culture (Supplementary Fig. 4). The cell-derived extracellular α-synuclein proteins that were used in this experiment contained both α-synuclein monomers and aggregates (Supplementary Fig. 5). Upon internalization, the monomers were collected mostly in the Triton X-100 soluble fractions, whereas the aggregates were found in the insoluble fractions (Fig. 3a, c). The cumulative levels of α-synuclein in Triton X-100 soluble fraction were not altered by overexpression of WT LRRK2 (Fig. 3a, b). However, WT LRRK2 overexpression brought an increase in accumulation of α-synuclein aggregates in the Triton-insoluble fraction (Fig. 3c, d). The expression of G2019S, the PD-linked mutant with increased kinase activity, further increased the accumulation of α-synuclein both in the Triton-soluble and -insoluble fractions in the recipient SH-SY5Y cells (Fig. 3a–d). In contrast, the expression of LRRK2 D1994A, a kinase-deficient variant, did not alter the accumulation of internalized α-synuclein (Fig. 3a–d).

Propagation of α-synuclein was quantitatively analyzed using a dual-cell BiFC system consisting of two different stable cell lines, which overexpressed either V1S or SV2[26]. Although both V1S- and SV2-expressing cells can secrete and internalize α-synuclein, our previous study showed that V1S cells secrete substantially more α-synuclein protein than SV2 cells, which makes the latter cells primary recipients in this assay system[26]. Transduction of SV2 recipient cells with WT LRRK2 resulted in an increase in the propagation of α-synuclein aggregates, while expression of the G2019S variant further increased the propagation to significantly higher levels than observed with WT LRRK2. These effects were not observed when the kinase-dead D1994A variant was expressed. (Fig. 3e, f, Supplementary Fig. 6). Next, we investigated whether the increase in LRRK2 kinase activity provokes the secretion of α-synuclein aggregates. Because α-synuclein is constantly secreted, internalized, and then secreted again[26], the

extracellular α-synuclein in the media is the sum of 'primary' secretion and the 'secondary' secretion (or internalization and resecretion). It is very difficult to measure only the 'primary' secretion'. However, we can measure the 'secondary secretion' by measuring the Venus fluorescence in the media, which represents co-aggregation between the internalized α-synuclein and the recipient-expressed α-synuclein. This secondary secretion was increased in the cultured cells overexpressing WT LRRK2 and even more increased in the cells expressing G2019S LRRK2 (Fig. 3g). For further validation of the role of LRRK2 kinase, we inhibited the LRRK2 kinase activity using the selective inhibitor HG-10-102-01[27]. Treatment of the cells with this inhibitor decreased LRRK2 autophosphorylation on several serine residues (Fig. 3h). Inhibition of LRRK2 kinase activity near completely nullified the effects of LRRK2 WT and G2019S on the propagation of α-synuclein (Fig. 3i, j). The inhibitor did not affect α-synuclein propagation in mock-transduced cells, probably due to the low expression levels of endogenous LRRK2 in SH-SY5Y cells. These results demonstrated that LRRK2 regulates α-synuclein propagation through its kinase activity.

**LRRK2-mediated RAB35 phosphorylation.** Next, we turned our attention to substrates of LRRK2 kinase. We were particularly interested in RAB proteins, small GTPases that regulate vesicle trafficking and sorting[28]. A recently published study showed that several RABs, including RAB3, 8, 10, 12, 35, and 43, were phosphorylated by LRRK2[29,30]. Our recent in vitro screen using protein chips identified three RAB proteins, RAB1, RAB8, and RAB35, as substrates of LRRK2 kinase[31]. All these RAB proteins contain a Thr residue in the Switch II domain. It has been shown that the Thr residue in this domain is the preferred substrate of LRRK2 kinase[29]. Substitution of Thr 72 with Ala abolished the LRRK2-mediated RAB35 phosphorylation[31]. These earlier studies encouraged us to identify the specific RAB proteins that are involved in LRRK2-mediated α-synuclein propagation. First, we determined which of these three RAB proteins co-localize with internalized α-synuclein in the recipient cells. RAB proteins were expressed using lentiviral vectors in the recipient cells. RAB1 and RAB35 exhibited a more diffuse pattern than RAB8 (Supplementary Fig. 7a), However,

vesicular patterns were present for all the RABs tested. We found that approximately 35% of the internalized α-synuclein colocalized with RAB35 (Fig. 4a, b and Supplementary Fig. 7b), whereas co-localization with RAB1 and RAB8 was less than 10% (Fig. 4a, b). Pearson's coefficient also showed a correlation between RAB35 and transmitted α-synuclein (Fig. 4c). Co-immunoprecipitation experiments confirmed that LRRK2 interacted with RAB35 in neuronal cells (Fig. 4d–g). We then examined the effects of ectopic expression

of a dominant-negative mutant (S22N) form of RAB35[32]. Inhibition of RAB35 via cell transduction with *RAB35* S22N completely nullified the LRRK2-stimulated α-synuclein propagation (Fig. 4h, i).

**RAB35 phosphorylation on α-synuclein propagation.** To further validate the role of RAB35 in α-synuclein propagation in vivo, BiFC transgenes (V1S and SV2) were introduced into WT

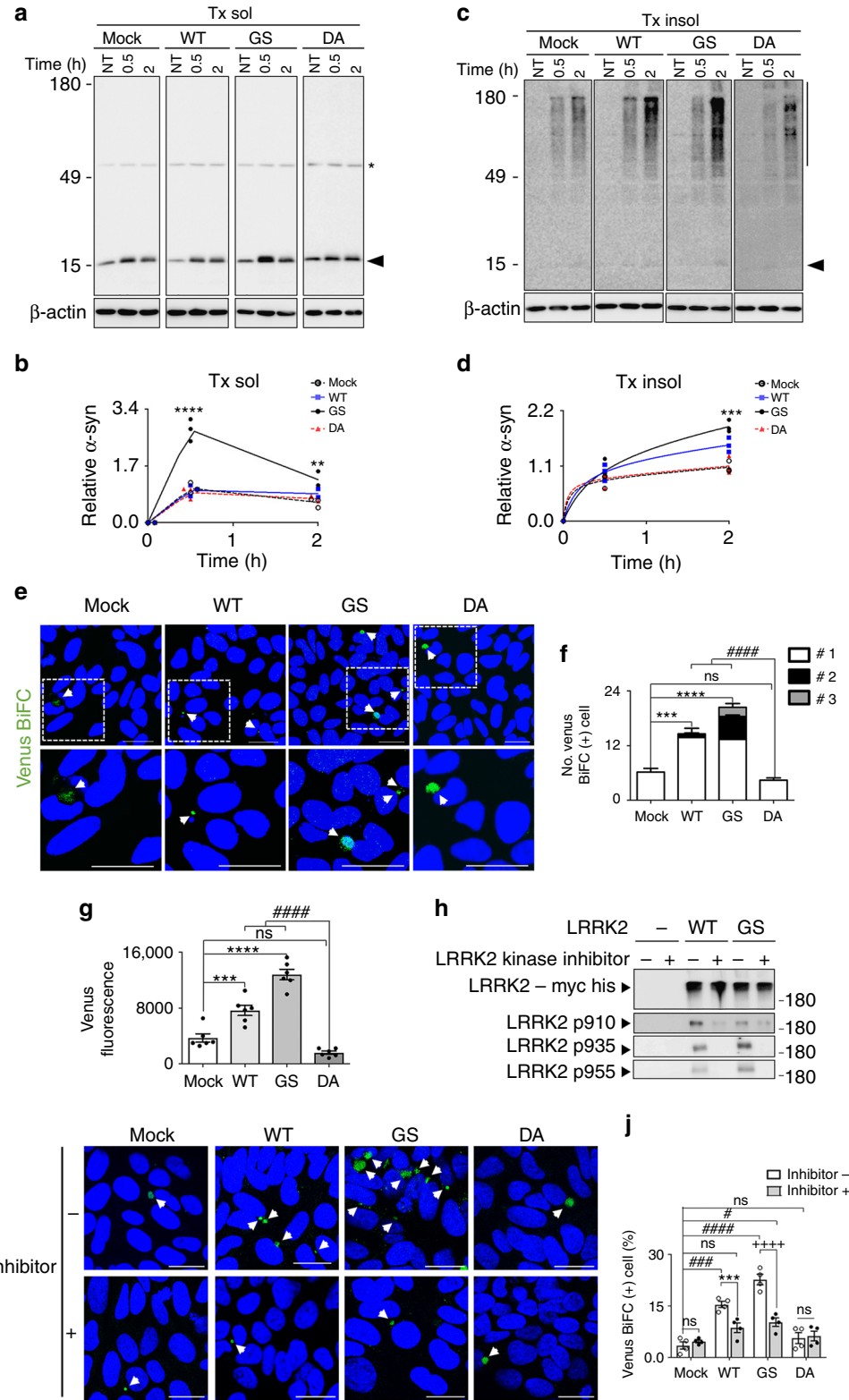

(N2) and *rab-35* mutant (*tm2058*) *C. elegans*, and three lines with similar α-synuclein expression levels were selected for analysis (Supplementary Fig. 8a, b). The *rab-35* mutant worms showed effects similar to those in *lrk-1* worms, exhibiting reduced levels of Venus fluorescence and Venus-positive inclusion numbers compared with the N2 control worms (Fig. 5a–c and Supplementary Fig. 8c–f). Consistent with the reduced aggregate propagation, the *rab-35* mutant worms showed reduced nerve degeneration, improved pharyngeal pumping, and increased life span (Fig. 5d–g and Supplementary Fig. 8g, h).

Next, we tried to reverse the phenotypes of *rab-35* mutant worms with ectopic expression of either the wild-type or T72A mutant form of human RAB35 protein; the latter is deficient in phosphorylation by LRRK2. We expressed either WT or T72A RAB35 transgenes in BiFC *C. elegans*, and the expression levels of human RAB35 were comparable (Supplementary Fig. 9a, b and d, e). Expression of the wild-type human RAB35 was able to recover the propagation of α-synuclein, whereas that of T72A RAB35 was not (Fig. 6a–c and Supplementary Fig. 9f, g and j, k). Likewise, nerve degeneration, pumping rate, and life span were significantly exacerbated in the wild-type RAB35-expressing worms, whereas the T72A RAB35-expressing worms did not show significant differences in these phenotypes compared with the *rab-35* mutant worm itself (Fig. 6d–g and Supplementary Fig. 9n, o).

We then examined whether expression of constitutively active RAB35 (Rab35 Q67L) can reverse the phenotypes of *lrk-1* mutant worms. We generated three independent RAB35 Q67L transgenic lines in the *lrk-1* mutant BiFC worms (Supplementary Fig. 9c–e). RAB35 Q67L worms reversed the *lrk-1* phenotypes, increasing the propagation of α-synuclein and inclusion formation (Fig. 7a–c and Supplementary Fig. 9h, i and l, m). These worms exacerbated the degenerative phenotypes, such as nerve degeneration, pharyngeal pumping behavior, and life span, compared with the *lrk-1* worms (Fig. 7d–g and Supplementary Fig. 9p, q). Collectively, these results suggest that phosphorylation of RAB35 on Thr72 and subsequent activation of this protein is the downstream effector of *lrk-1* in the regulation of α-synuclein propagation.

**Effect of an LRRK2 inhibitor on α-synuclein pathology**. To investigate the role of LRRK2 kinase activity in α-synuclein deposition, we administered an LRRK2 inhibitor (HG-10-102-01, 10 mg kg$^{-1}$) via intraperitoneal (IP) injection to either non-transgenic (non-tg) or α-synuclein tg mice for 4 weeks. Inhibition

of LRRK2 kinase activity was confirmed by measuring LRRK2 autophosphorylation in the brain homogenates, which showed a significant 70% reduction in non-tg and α-synuclein tg mice treated with HG-10-102-01 compared with vehicle (Supplementary Fig. 10a, e). Levels of total LRRK2 were not different among the four groups (Supplementary Fig. 10a, d). Immunohistochemical analysis demonstrated that in the vehicle-treated α-synuclein tg mice, there was abnormal accumulation of α-synuclein aggregates in the neocortex (Fig. 8a–c), striatum (Fig. 8a, d), and corpus callosum (Fig. 8a, e). Moreover, α-synuclein was highly accumulated in neuronal cell bodies (Fig. 8a–c), neuropils (Fig. 8a, b–e), and axons (Fig. 8a–d) of α-synuclein tg mice. The α-synuclein aggregates were highly phosphorylated (pSer129) (Fig. 8f–h) and showed proteinase-K (PK) resistance (Fig. 8f, i, j). Administration of the LRRK2 inhibitor significantly decreased the levels of α-synuclein aggregates in α-synuclein tg mice in the neocortex, striatum and corpus callosum (Fig. 8a–e). The levels of phosphorylated α-synuclein in the corpus callosum (Fig. 8f, h) and PK-resistant aggregates in the neocortex and corpus callosum (Fig. 8f, i, j) were also decreased by LRRK2 inhibitor treatment. In particular, administration of the LRRK2 inhibitor significantly reduced trans-axonal α-synuclein aggregates, a possible mechanism for α-synuclein propagation[33]. Immunoblot analysis also showed that in the vehicle-treated α-synuclein tg mice there was an accumulation of both α-synuclein monomers and high molecular weight aggregates, which were clearly decreased by treatment with the LRRK2 inhibitor (Supplementary Fig. 10a, b). Taken together, these results support the notion that functional inhibition of LRRK2 kinase activity ameliorates α-synuclein pathology in a mouse model of PD.

**LRRK2 modulates α-synuclein trafficking**. To investigate the mechanisms through which inhibition of LRRK2 might reduce α-synuclein pathology, we next analyzed the levels of RAB35 and their relationship with α-synuclein and lysosomal markers. By western blot, the levels of RAB35 were increased in vehicle-treated α-synuclein tg mouse brains, whereas treatment with the LRRK2 kinase inhibitor reduced RAB35 in tg mice to levels comparable to those in non-tg controls (Supplementary Fig. 10a, c). Our attempts to measure the levels of phospho-RAB35 were unsuccessful, which needs to be addressed in the future. Similar to the western blot results, immunocytochemical analysis demonstrated that the levels of RAB35 were increased in the neocortex

**Fig. 3** LRRK2 kinase activity regulates propagation of α-synuclein. **a-d** The accumulation of internalized α-synuclein in Triton X-100 soluble fraction (Tx sol) (**a, b**) and Triton X-100 insoluble fraction (Tx insol) (**c, d**). The α-synuclein monoclonal antibody Syn-1 from BD Biosciences was used for detection of α-synuclein. NT represents non-treated control. Arrowheads in **a** and **c** indicate α-synuclein monomer. Bar on the right side of blot in **c** represents the α-synuclein aggregates. Asterisk in **a** represents non-specific binding of antibody. Quantified regions were indicated on the right as an arrowhead in **a** and a line in (**c**). Black open circle: Mock, blue closed square: LRRK2 WT, black closed circle: LRRK2 G2019S (GS), red closed triangle: LRRK2 D1994A (DA). **b, d** Relative levels of internalized α-synuclein in Tx sol (**b**) and Tx insol (**d**). **b** $N = 3$, Transgene $F_{(3,24)} = 60.79$, Time $F_{(2,24)} = 264.8$, Interaction $F_{(6,24)} = 28.42$, ** $p < 0.01$, **** $p < 0.001$. **d** $N = 3$, Transgene $F_{(3,24)} = 13.23$, Time $F_{(2,24)} = 352.2$, Interaction $F_{(6,24)} = 7.237$, * $p < 0.05$, *** $p < 0.005$, Graphs in **b, d** were analyzed by two-way ANOVA with Dunnet's post hoc test. **e, f** Alterations in LRRK2 kinase activity regulate propagation of α-synuclein. **e** The effects of LRRK2 kinase activity on the propagation of α-synuclein. Arrowhead: Venus BiFC puncta. Blue: Nuclei. Scale: 20 μm. **f** The number of Venus BiFC (+) cell. The color bars in graph (**f**) represent the number of Venus BiFC puncta in Venus BiFC fluorescence (+) cells. Five independent experiments were performed. Two hundred cells were analyzed per each experiment. Y axis represents the number of Venus BiFC(+) cells out of 200 cells analyzed. Transgene $F_{(3,48)} = 3.09$, Puncta number $F_{(2,48)} = 172.1$, Interaction $F_{(6,48)} = 10.4$, ns: not significant, *** $p < 0.005$, **** $p < 0.001$, #### $p < 0.001$ by two-way ANOVA with Tukey's post hoc test. **g** Effects of LRRK2 kinase activity on the secondary secretion of Venus (+) α-synuclein aggregates. $N = 6$, $F_{(3,20)} = 44.77$, ns: not significant, *** $p < 0.005$, **** $p < 0.001$, #### $p < 0.001$ by one-way ANOVA with Tukey's post hoc test. **h–j** Pharmacological inhibition of LRRK2 kinase activity diminishes LRRK2-induced α-synuclein propagation. **h** The autophosphorylation of LRRK2. **i** The propagation of α-synuclein. Arrowhead: Venus BiFC puncta. Blue: Nuclei. Scale: 20 μm. **j** Quantification of Venus BiFC fluorescence (+) cell (%). Four independent experiments were performed. Two hundred cells were analyzed per each experiment. Treatment $F_{(1,24)} = 24.12$, Transgene $F_{(3,24)} = 41.28$, Interaction $F_{(3,24)} = 13.35$, ns: not significant, *** $p < 0.005$, ++++ $p < 0.0001$, # $p < 0.05$, ### $p < 0.005$, #### $p < 0.0001$ by two-way ANOVA with Tukey's post hoc test. Data are represented as mean ± SEM

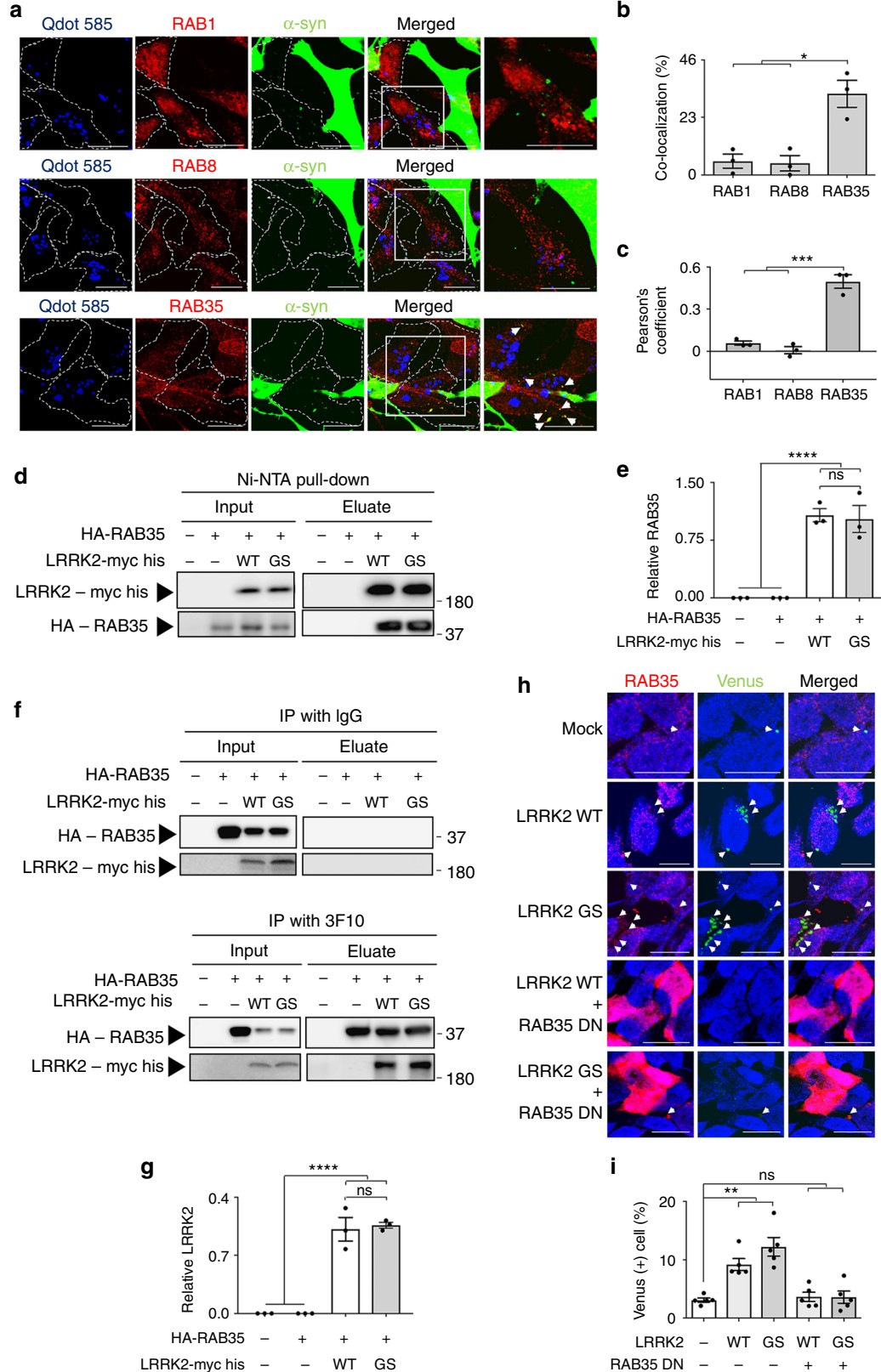

of vehicle-treated α-synuclein tg mice (Fig. 9a–c). The confocal imaging analysis showed that RAB35 was localized to punctate intraneuronal structures consistent with endosomes; in the vehicle- and LRRK2 inhibitor-treated non-tg mice, these granular structures measured on average 0.942±0.08 and 0.903±0.05 μm, respectively, while in the vehicle-treated α-synuclein tg mice, they

measured 1.707±0.19 μm. Treatment with the LRRK2 inhibitor normalized the size of the RAB35-positive endosomes (0.935 ±0.07 μm) in the α-synuclein tg mice. Next, we analyzed co-localization of α-synuclein to the RAB35-positive endosomes. This analysis showed minimal co-localization of α-synuclein with the RAB35 structures in the neuronal cell bodies in the neocortex

**Fig. 4** RAB35 mediates LRRK2-induced propagation of α-synuclein. **a-c** Localization of transferred α-synuclein in the co-culture of α-synuclein overexpressing SH-SY5Y cells (donor cells) and naive SH-SY5Y cells (recipient cells labeled with Q tracker). **a** The localization of transmitted α-synuclein was analyzed by co-immunostaining with RAB proteins (RAB1, RAB8, and RAB35). Red: RAB proteins, Blue: Qdot 585, recipient cell marker, arrowhead: transmitted α-synuclein, colocalized with RAB proteins, scale bar: 20 μm. The percentage of RAB (+) transmitted α-synuclein was quantified in **b**. Three independent experiments were performed. Three hundred cells were analyzed per each experiment. F(2,6)=16.09, * $p < 0.05$ by one-way ANOVA with Dunnet's post hoc test. The pearson's coefficient was calculated in (**c**). Three independent experiments were performed. Fifty cells were analyzed per each experiment. F(2,6)=68.1, *** $p < 0.005$ determined by one-way ANOVA with Dunnet's post hoc test. **d-g** Interaction between LRRK2 and RAB35. WT: LRRK2 WT, GS: LRRK2 G2019S. **d** Pull-down assay of 6X histidine conjugated LRRK2 with nickel beads. The relative levels of eluted RAB35 were calculated in **e**. N = 3, F(3,8)=39.08, ns: not significant, **** $p < 0.0001$ by one-way ANOVA with Tukey's post hoc test. **f** Immunoprecipitation of RAB35 with 3F10, monoclonal antibody against HA tag. Mouse IgG was used as a control. The relative levels of eluted LRRK2 were quantified in **g**. N = 3, F(3,8)=66.84, ns: not significant, **** $p < 0.0001$ by one-way ANOVA with Tukey's post hoc test. **h, i** Dominant-negative mutant RAB35 S22N (RAB35 DN) ameliorated LRRK2-induced-propagation of α-synuclein. Propagation of α-synuclein aggregates were calculated by measuring the Venus BiFC (+) cell (%). Arrowhead: Venus BiFC (+) puncta, red: RAB35, blue: nuclei, scale: 20 μm. The percentage of Venus BiFC (+) cells was calculated in (**i**). Five independent experiments were performed. Three hundred cells were analyzed per each experiment. F(4,20)=15.79, ns: non-significant, ** $p < 0.01$ by one-way ANOVA with Dunnet's post hoc test. Data are represented as mean ± SEM

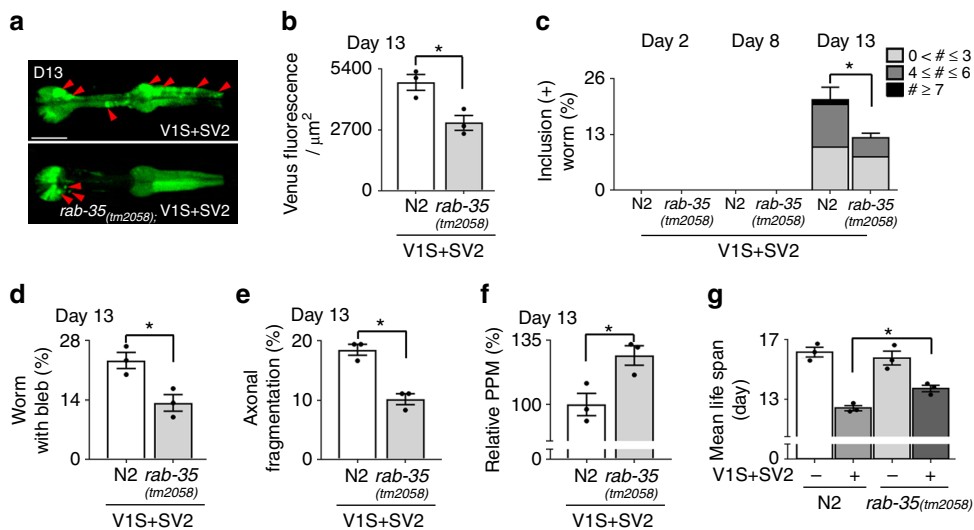

**Fig. 5** Mutation in *rab-35* resulted in the similar phenotypes as *lrk-1*. **a, b** Venus BiFC fluorescence in wild-type and *rab-35* transgenic models at day 13. The red arrowheads point to Venus BiFC-positive inclusions in pharynx. Twenty worms for each line were used (N = 3 in each group), * $p < 0.05$, Scale bars: 200 μm. **c** Worms with Venus BiFC-positive inclusions were quantified at the different ages. The color bars in graph (**c**) indicate percentages of Venus BiFC-positive inclusions. Thirty worms for each line were used (N = 3 in each group), * $P < 0.05$. **d** Quantification of worms that have axonal blebs. **e** Nerve fragmentation of each transgenic line at day 13. **f** Relative pharyngeal pumping rates at day 13. Thirty worms for each line were used (N = 3 in each group), * $p < 0.05$. All error bars represent SEM, P value, * $p < 0.05$, was measured by unpaired, two-tailed Student's *t* test. **g** Mean life span. One hundred worms for each line were used (N = 3 in each group). F(3, 9)=33.730, P value,* $p < 0.05$, was calculated by one-way ANOVA with Tukey's post hoc test. All values are represented as mean ± SEM

of the non-tg mice (vehicle and LRRK2 inhibitor). In contrast, the vehicle-treated α-synuclein tg mice displayed a significant increase in co-localization between these two markers, and treatment with the LRRK2 inhibitor reduced the % co-localization in the α-synuclein tg mice (Fig. 9a, b). Finally, we evaluated whether the effects of the LRRK2 inhibitor on RAB35 resulted in α-synuclein clearance via lysosomes. For this purpose, confocal analysis was performed in sections double labeled with α-synuclein and cathepsin D. This analysis showed a low % of co-localization between α-synuclein and cathepsin D structures in neuronal cell bodies in the neocortex of the non-tg mice (vehicle and LRRK2 inhibitor). In contrast, the vehicle-treated α-synuclein tg mice showed a moderate increase in co-localization between these two markers, and treatment with the LRRK2 inhibitor significantly increased the % co-localization in the α-synuclein tg mice (Fig. 9d, e). These results suggest that while under pathological conditions α-synuclein localized to RAB35-positive compartments when LRRK2 was activated, inhibition of LRRK2 kinase resulted in reduced localization of α-synuclein in RAB35-

positive compartments and increased trafficking to the lysosome (cathepsin D), which might facilitate α-synuclein clearance.

## Discussion

Here, we show that LRRK2 plays a critical role in propagation of α-synuclein. The effects of LRRK2 on α-synuclein propagation require the LRRK2 kinase activity. LRRK2 promotes α-synuclein propagation by phosphorylating RAB35. Administration of a selective LRRK2 kinase inhibitor slowed the propagation of α-synuclein in culture and reduced synucleinopathy lesions in a transgenic mouse model.

Propagation of α-synuclein pathology was markedly reduced as a result of *lrk-1* deletion in *C. elegans* and *Lrrk2* ablation in rats. The similarity of the findings in these two in vivo models strengthens the likelihood of a relationship between LRRK2 and LRK-1 functions. However, interpretation of the in vivo work assessing α-synuclein propagation requires a few notes of caution. The WT and $Lrrk2^{-/-}$ rats used for these studies were of the

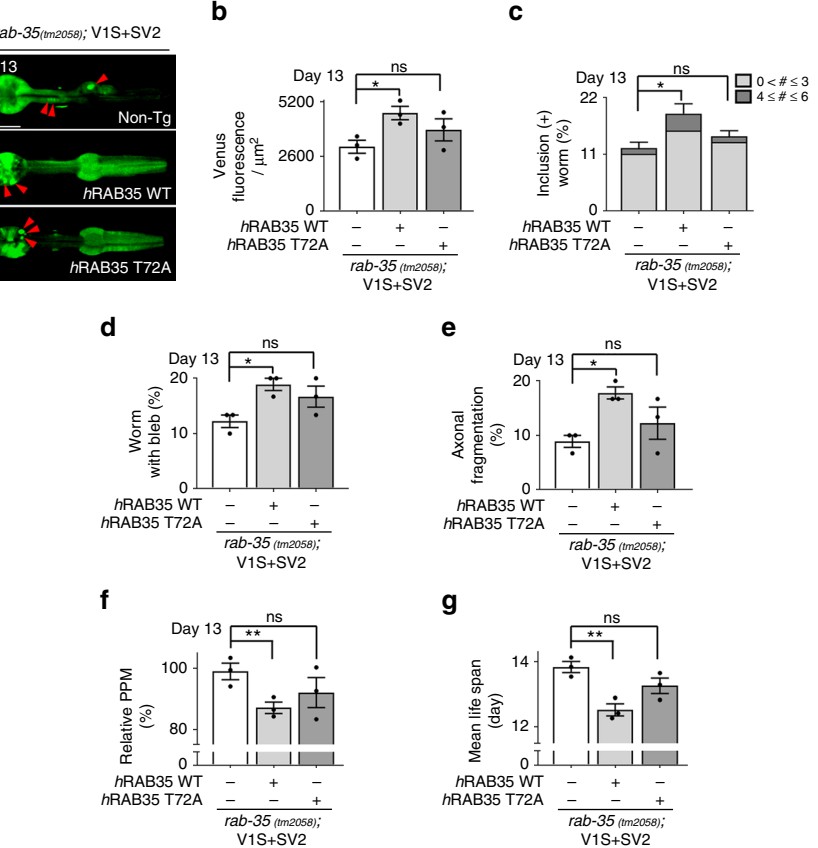

**Fig. 6** RAB35 Phosphorylation is necessary for regulation of α-synuclein propagation. **a** Venus BiFC fluorescence in Non-Transgenic (Non-Tg), *h*RAB35 WT, and *h*RAB35 T72A transgenic line at day 13. The red arrowheads indicate Venus BiFC-positive inclusions in pharynx. Scale bars: 200 μm. **b** Quantification of Venus BiFC fluorescence at day 13. Thirty worms for each line were used ($N = 3$ in each group). $F_{(2, 6)} = 4.105$, ns: not significant, * $p < 0.05$. **c** Worms with Venus BiFC-positive inclusions at day 13. The color bars in graph (**c**) show percentages of worms that have Venus BiFC-positive inclusions in *rab-35 (tm2058)* BiFC transgenic background. Thirty worms for each line were used ($N = 3$ in each group). $F_{(2, 6)} = 4.670$, ns: not significant, * $p < 0.05$. **d, e** Quantification of axonal bleb number (**d**) and fragmentation (**e**) at day 13. Thirty worms for each line were used ($N = 3$ in each group). $F_{(2, 6)} = 5.607$ (**d**), $F_{(2, 6)} = 5.450$ (**e**), ns: not significant, * $p < 0.05$. **f** Relative pharyngeal pumping rates at day 13. Thirty worms for each line were used ($N = 3$ in each group). $F_{(2, 180)} = 4.421$, ns: not significant, ** $p < 0.01$. **g** Mean life span ($N = 3$ in each group). One hundred worms for each line were used ($N = 3$ in each group). $F_{(2, 6)} = 10.750$, ns: not significant, ** $p < 0.01$. All values shown in figures are represented as mean ± SEM. *P* values, including ns: not significant, * $p < 0.05$, ** $p < 0.01$, were calculated by one-way ANOVA with Dunnet's post hoc test

same genetic background, albeit not littermates; this raises the possibility that other genetic variations besides the lack of *Lrrk2* might have contributed to the reduction in α-synuclein spreading in deficient animals. In *C. elegans*, the *lrk-1* gene has an ancient origin distinct from that of *LRRK2*, while perhaps representing a conserved homologue of mammalian *LRRK1*[34]. LRRK1 is important during embryogenesis and development in mammals, but absent very soon after birth[35]. The relationships between *lrk-1* and *LRRK2*, which is expressed exclusively in adult mammalian tissues, are not clear. For example, both LRK-1 and LRRK1 lack the N-terminal region that is present in LRRK2. However, the consistency in the effects of *lrk-1* and mammalian *Lrrk2* on α-synuclein propagation observed in the present study suggests a functional overlap between these genes. The fact that both *lrk-1* and *Lrrk2* regulate α-synuclein propagation also indicates that the N-terminal region is not required for this function. However, it is possible that the N-terminal region may have functions other than α-synuclein propagation, e.g., regional and cell type vulnerability. In this regard, it is interesting to note that most of the PD-linked mutations are located outside the N-terminal region in LRRK2[36]. Another aspect of *lrk-1* to consider when interpreting the worm data is that the worm *lrk-1* mutants have several phenotypes, such as abnormal axonal arborizations, synaptic

dysfunction, and mislocalization of axonal proteins in dendrites[37–42]. Many of these alterations might be due to disturbed neuronal development or general worm health in the absence of a LRRK1/2 homologue and might impact α-synuclein propagation indirectly. In the present study, we did not analyze the previously described phenotypes of *lrk-1* mutant worms nor did we analyze these phenotypes in *rab-35* mutant worms. Therefore, it is not clear whether *rab-35* is involved in the *lrk-1* phenotypes other than its role in α-synuclein propagation.

In a recent study by Tyson et al.[23], the role of LRK-1 in α-synuclein propagation was investigated in a *C. elegans* model. This study reported that silencing of *lrk-1* expression by approximately 20% increased α-synuclein propagation, whereas in our current study, where the *lrk-1* null mutant worm was used, the opposite result was observed, specifically, deletion of the *lrk-1* gene enhanced the propagation. The source of this discrepancy is not known. There are several differences in experimental design between these studies, which may account for the different results. The former study employed EGFP-BiFC, whereas our study used Venus BiFC. In addition, Tyson et al. designed their worm model in such a way that they could monitor neuron-to-neuron propagation, while our model allows monitoring of propagation between neurons and muscle cells. Perhaps the most

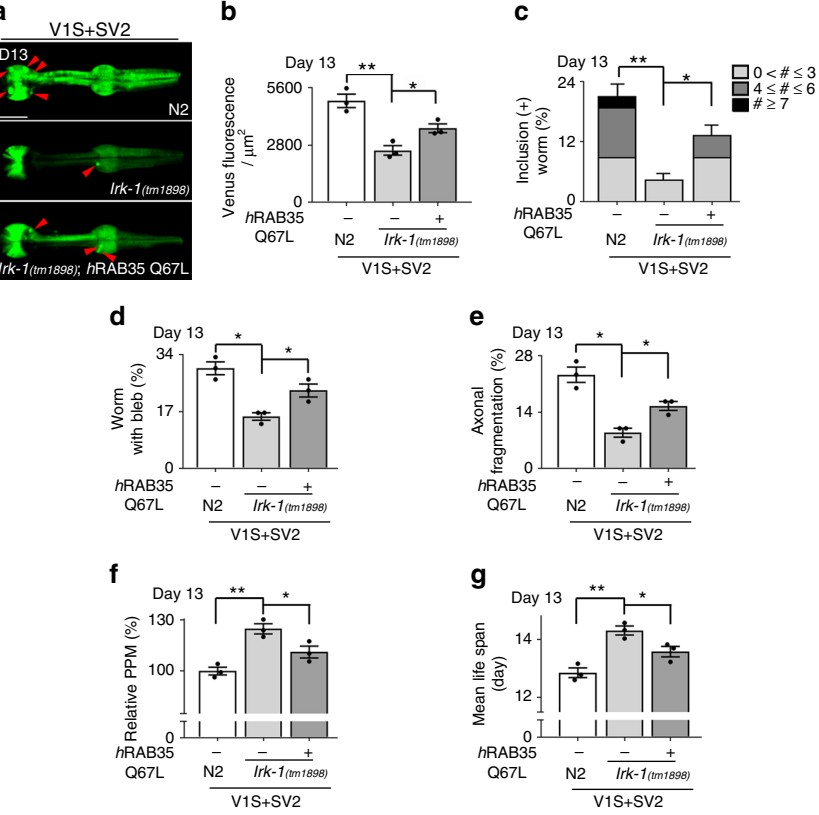

**Fig. 7** RAB35 mediates α-synuclein propagation in the downstream of *lrk-1*. **a, b** Venus BiFC fluorescence in wild-type, *lrk-1* background worm, and *h*RAB35 Q67 L transgenic in *lrk-1* background worm at day 13. The red arrowheads designate inclusions in pharynx. Thirty worms for each line were used (*N* = 3 in each group). F(2, 6)= 21.00, * *p* < 0.05, ** *p* < 0.01. Scale bars: 200 μm. **c** Venus BiFC-positive inclusions at day 13. The color bars in graph (**c**) display the percentages of Venus BiFC-positive inclusions (+) worms. Thirty worms for each line were used (*N* = 3 in each group). F(2, 6)=21.13, * *p* < 0.05, ** *p* < 0.01. **d, e** Degeneration of axonal processes. The percentage of worms with neuritic blebs (**d**) and fragmentation of axonal processes (**e**) at day 13. Thirty worms for each line were used (*N* = 3 in each group). F(2, 6)=18.140, (**d**), F(2, 6)= 25.370 (**e**), * *p* < 0.05. **f** Relative pharyngeal pumping rates at day 13 (*N* = 3 in each group). F(2, 180)=13.670, * *p* < 0.05, ** *p* < 0.01. The transgenic lines expressing constitutively active form showed more severe behavioral defects than *lrk-1* transgenic worms. **g** Mean life span. One hundred worms for each line were used (*N* = 3 in each group. The overexpression of transgene under the deficiency of *lrk-1* worsened disease phenotype. F(2, 6)=18.910, * *p* < 0.05, ** *p* < 0.01. All values represent the mean ± SEM. *P* values, including * *p* < 0.05, ** *p* < 0.01, were calculated by one-way ANOVA with Dunnet's post hoc test

important difference is the way and the extent of the manipulation of *lrk-1* expression. To reduce *lrk-1* expression, Tyson et al. used RNAi, resulting in a reduction of only 20–30% according to mRNA measurement. In contrast, we used mutant worms with *lrk-1* deletion that completely eliminated gene expression. Therefore, it is possible that due to a dose effect and/or differences in compensatory responses, a small decrease in *lrk-1* expression yields a different phenotype than that produced by complete *lrk-1* depletion. It is also noteworthy that total elimination of the *Lrrk2* gene in rats resulted in an effect similar to that observed in *lrk-1* mutant worms, suggesting that the propagation phenotype of *lrk-1* deletion is conserved in mammals and underscoring a direct relationship between the absence of *lrk-1* or *Lrrk2* and inhibition of α-synuclein spreading.

In worm models, inclusion bodies and other degenerative phenotypes were analyzed on day 13, at which time these phenotypes are well expressed. It needs to be noted that about half the worms were already dead by day 13. Therefore, some of the quantitation values, such as 20% worms with inclusion bodies, may be underestimated because a number of the dead worms may have had inclusions before death. In the present study, the relationships among inclusion bodies, axonopathies, and mortality were not clearly addressed. The inclusion bodies may be an independent phenotype from axonal pathology and increased

mortality because the worms with inclusions were less frequent than those with the axonal phenotype.

In the worm and SH-SY5Y BiFC models, it is not clear which species are transmitted from one cell to another. The data can be explained by V1S monomers being secreted by one group of cells in the co-culture and then being taken up by cells expressing SV2 monomers, or vice versa. Additionally, one cell type could release V1S monomers and the other cells in the co-culture could release SV2 monomers, resulting in the generation of Venus fluorescence in the medium from the V1S/SV2 dimer. The latter possibility of monomers forming dimers in the medium was ruled out in an experiment in which the culture media from V1S and SV2 cells were separately collected and mixed. This experiment did not result in Venus fluorescence (Supplementary Fig. 11).

In cells and animal models, LRRK2 has been shown to function in vesicle trafficking through the endolysosomal and autophagic pathways[43–46]. Defects in this function might lead to a reduced number of neuritic processes in developing neurons[47]. α-Synuclein aggregates secreted from neuronal cells have been shown to be internalized into other neurons through endocytosis. These internalized aggregates were transported through the endosomal pathway and eventually wound up in lysosomes for destruction[6]. Therefore, it has been postulated that perturbation of this pathway interferes with the degradation of internalized α-

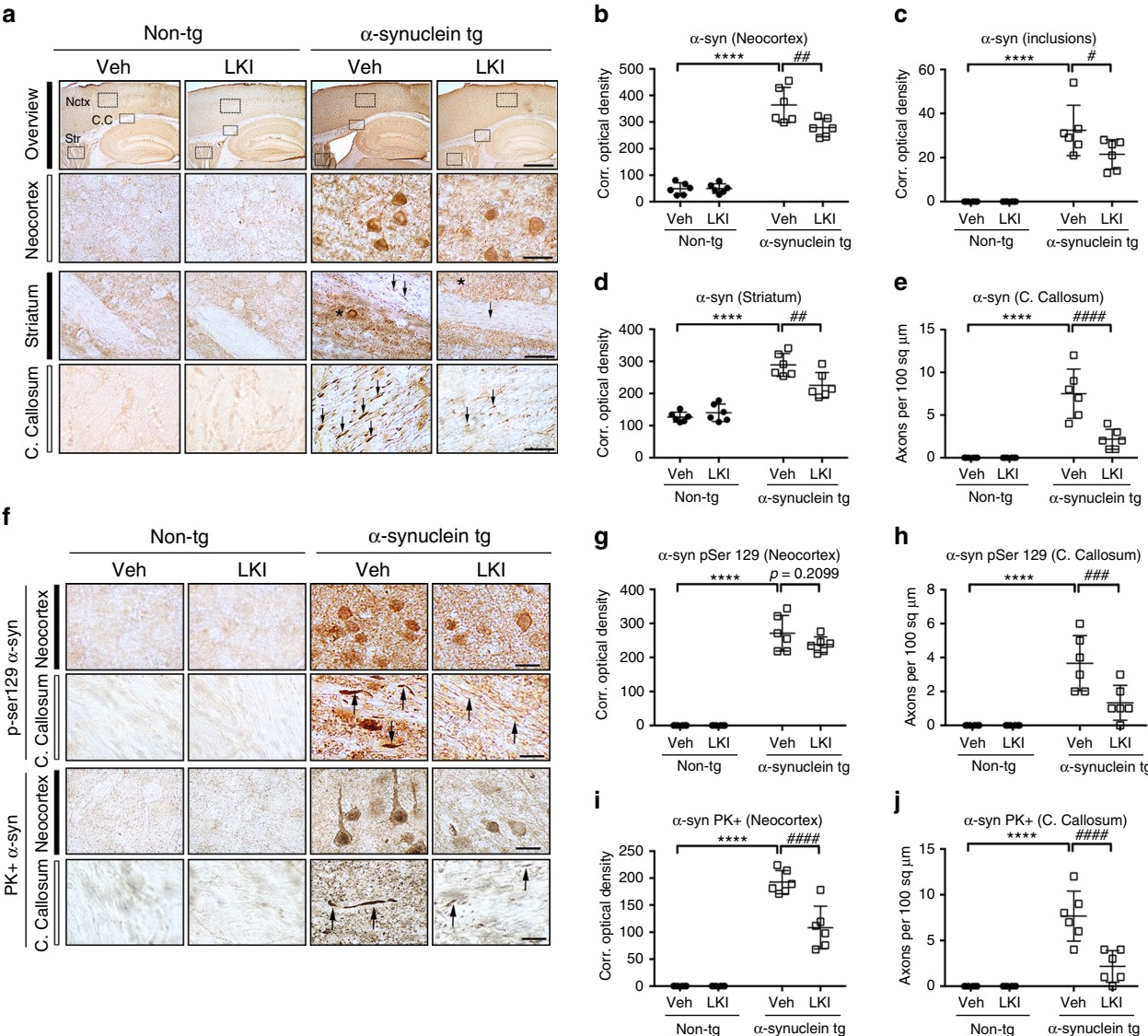

**Fig. 8** Blockade of LRRK2 kinase inhibits α–synuclein propagation in mice. **a** Representative immunohistochemical analysis of α-synuclein in the neocortex (Nctx), striatum (Str), and corpus callosum (C.C) of non-transgenic (non-tg) and α-syn tg mice. Age-matched mice were injected with either vehicle (Veh) or LRRK2 inhibitor (LKI) for 4 weeks. Scale bars, 250 μm (low magnification) and 25 μm (high magnification). **b–e** Optical analysis of α-synuclein deposition in non-tg and α-syn tg mice (**b**) Optical density analysis of α-synuclein immunoreactivity in neocortex. Treatment $F_{(1,20)}$=6.601, Genotype $F_{(1,20)}$=276.8, Interaction $F_{(1,20)}$=6.919, **** $p < 0.001$, ## $p < 0.01$. **c** The number of α-synuclein inclusion positive cells in neocortex. Treatment $F_{(1,20)}$=4.034, Genotype $F_{(1,20)}$=99.61, Interaction $F_{(1,20)}$=4.034, **** $p < 0.001$, # $p < 0.05$. **d** Optical density analysis of α-synuclein immunoreactivity in striatum. Treatment $F_{(1,20)}$=3.88, Genotype $F_{(1,20)}$=96.22, Interaction $F_{(1,20)}$=9.303, **** $p < 0.001$, ## $p < 0.01$. **e** The number of α-synuclein-positive axons in corpus callosum. Treatment $F_{(1,20)}$=17.66, Genotype $F_{(1,20)}$=58, Interaction $F_{(1,20)}$=17.66, **** $p < 0.001$, #### $p < 0.001$. **f** Representative immunohistochemical analysis of phosphorylated (pSer129)- and Proteinase K (PK)-resistant α-synuclein in the neocortex and corpus callosum of non-tg and α-syn tg mice. Brain sections were immunostained against either phosphorylated- α-synuclein (pSer129) or α-synuclein following PK treatment. Scale bars, 250 μm (low magnification) and 25 μm (high magnification). **g** Optical density analysis of phosphorylated α-synuclein immunoreactivity in neocortex. Treatment $F_{(1,20)}$=2.064, Genotype $F_{(1,20)}$=476.2, Interaction $F_{(1,20)}$=2.064, **** $p < 0.001$. **h** The number of phosphorylated α-synuclein-positive axons in corpus callosum. Treatment $F_{(1,20)}$=8.75, Genotype $F_{(1,20)}$=40.18, Interaction $F_{(1,20)}$=8.75, **** $p < 0.001$, ### $p < 0.005$. **i** Optical density analysis of PK-resistant α-synuclein immunoreactivity in neocortex. Treatment $F_{(1,20)}$=20.97, Genotype $F_{(1,20)}$=269.3, Interaction $F_{(1,20)}$=20.97, **** $p < 0.001$, #### $p < 0.001$. **j** The number of PK-resistant α-synuclein-positive axons in corpus callosum. Treatment $F_{(1,20)}$=17.4, Genotype $F_{(1,20)}$=55.61, Interaction $F_{(1,20)}$=17.4, **** $p < 0.001$, #### $p < 0.001$. All analyses were done with $N = 6$ mice per group. All data were analyzed by using two-way ANOVA with Tukey's post hoc test

synuclein, thereby inducing aggregation of endogenous α-synuclein protein. Defects in the endolysosomal pathway indeed caused an increase in α-synuclein propagation. Mutations in lysosomal hydrolases, such as GBA1 and cathepsins, augmented the propagation[8,26,48]. Furthermore, endosomal trafficking defects through lowered expression of CHMP2B, a component of the ESCRT III complex, reduced lysosomal degradation of

internalized α-synuclein and increased propagation of α-synuclein[49]. These results suggest that diversion from endosomal trafficking and subsequent lysosomal degradation are the pivotal events in the propagation of α-synuclein. Interestingly, expression of mutant LRRK2 caused endolysosomal defects, leading to accumulation of abnormal lysosomal organelles[50]. Expression of mutant LRRK2 also disturbed RAB7-dependent

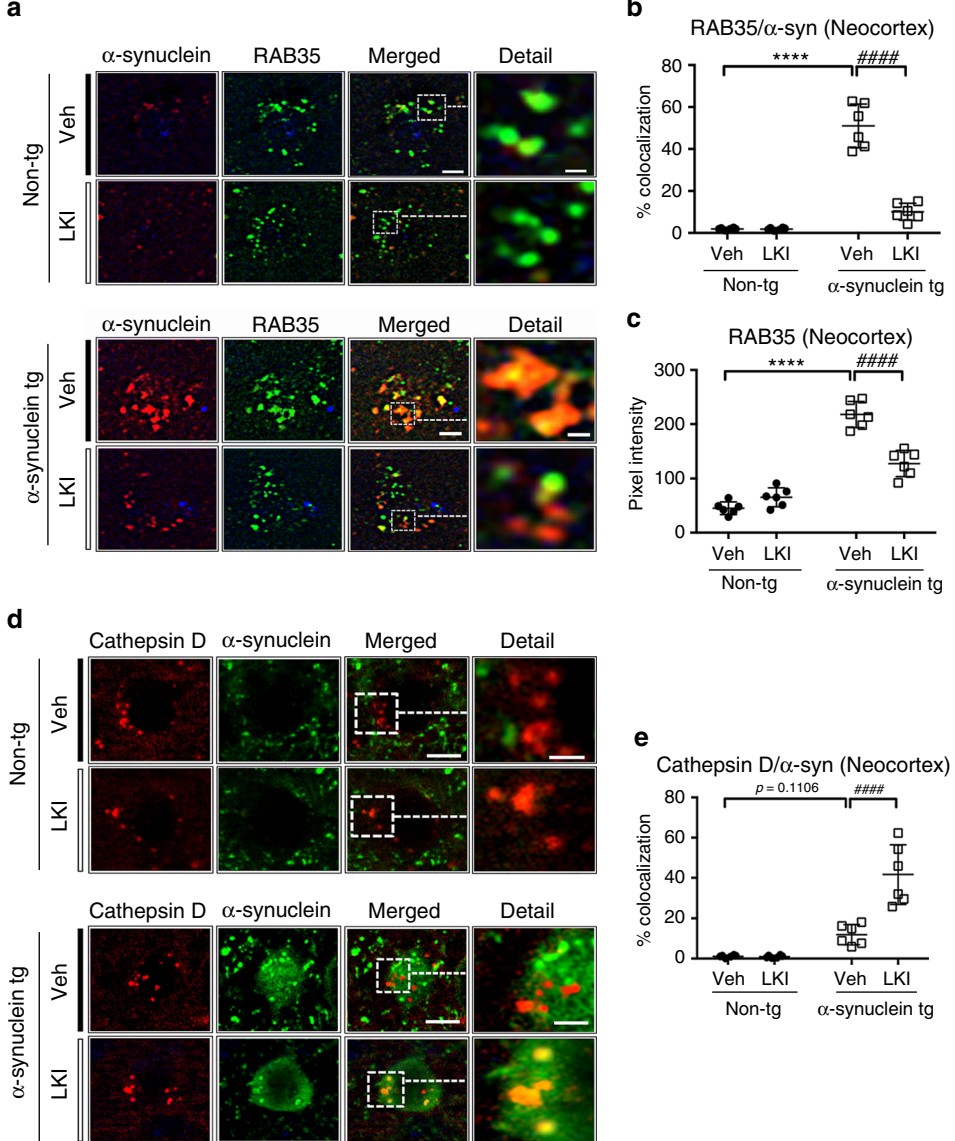

**Fig. 9** Blockade of LRRK2 kinase regulates traffkicking of α–synuclein in mice. **a** Non-tg and α-syn tg mice were treated with either vehicle or LRRK2 inhibitor for 4 weeks. Brain sections were double-immunolabelled with antibodies against α-synuclein and RAB35. Scale bars, 25 μm. **b** Co-localization analysis of α-synuclein and RAB35 in neocortex of non-tg and α-syn-tg mice. Treatment $F_{(1,20)}=79.8$, Genotype $F_{(1,20)}=156$, Interaction $F_{(1,20)}=79.15$, **** $p < 0.001$, #### $p < 0.001$. **c** Fluorescence intensity analysis for RAB35 in neocortex of non-tg and α-syn tg mice. Treatment $F_{(1,20)}=18.58$, Genotype $F_{(1,20)}=208.1$, Interaction $F_{(1,20)}=46.07$, **** $p < 0.001$, #### $p < 0.001$. **d** Non-tg and α-syn tg mice were administrated with either vehicle or LRRK2 inhibitor for 4 weeks. Brain sections were double-immunolabelled with antibodies against α-synuclein and Cathepsin-D. Scale bars, 25 μm. **e** Co-localization analysis of α-synuclein and Cathepsin D in neocortex of non-tg and α-syn-tg mice. Treatment $F_{(1,20)}=21.55$, Genotype $F_{(1,20)}=65.25$, Interaction $F_{(1,20)}=22.01$, #### $p < 0.001$. All data were analyzed by using two-way ANOVA with Tukey's post hoc test. All analysis was done with $N = 6$ mice per group. Data are represented as mean ± SEM

late endosome maturation, resulting in impaired lysosomal degradation of endocytosed receptors[51]. Therefore, LRRK2 might regulate α-synuclein propagation by perturbing endolysosomal functions, shunting the internalized α-synuclein aggregates from lysosomal degradation in recipient cells.

LRRK2 is a large multidomain protein containing a kinase domain, a GTPase domain, and several protein-interaction domains[52]. Several researchers have searched for LRRK2-interacting proteins and the substrates of LRRK2 kinase under the assumption that these proteins might act as effectors of LRRK2 functions[29,53]. In agreement with its functions in vesicle trafficking, LRRK2 has been shown to interact with RAB7L1 (RAB29)[54], a member of the RAB protein family that is involved in selective sorting of vesicles in the transport process[55,56]. The

search for LRRK2 kinase substrates identified several RAB proteins, including RAB3A, RAB8A, RAB10, and RAB12[29]. Our in vitro substrate screen identified RAB1 and RAB35 in addition to RAB3 and RAB8[31]. Among these RAB proteins, we demonstrated that RAB35 mediates the function of LRRK2 in α-synuclein propagation.

How RAB35 activation regulates α-synuclein propagation is unclear. RAB35 plays important roles in endosomal recycling and actin filament remodeling[57,58]. Through these functions, RAB35 has been recognized as the regulator of neurite outgrowth[59], cytokinesis[60], phagocytosis[61], cell adhesion[62], cell migration[63], cell polarity[64], and exosome secretion[65]. Considering its role in endosomal recycling, it is tempting to speculate that activation of the LRRK2-RAB35 pathway would hijack the internalized α-

synuclein aggregates from the lysosomal degradation pathway, leading to amplification of aggregates and continuous propagation (Supplementary Fig. 12). Therefore, excessive activation of RAB35 might be pathogenic by causing impaired homeostasis in endosomal trafficking of α-synuclein aggregates, particularly at the crossroads between lysosomal degradation and endosomal recycling. It is interesting to note that RAB35 expression is elevated in the serum of PD patients and in brain tissues of mouse models of PD, including LRRK2 G2019S transgenic mice[66]. In a neuroblastoma cell model, overexpression of RAB35 led to increased aggregation and secretion of α-synuclein A53T[66].

Our results showed that the LRRK2 kinase inhibitor reduced LRRK2 autophosphorylation but also the RAB35 and α-synuclein levels in mouse brain. This may be the result of the rescue effects of the LRRK2 inhibitor. Given our proposal that RAB35 phosphorylation is a downstream effect of LRRK2, direct assessment of the role of RAB35 phosphorylation in α-synuclein propagation in mice becomes important. The reduction in RAB35 levels could be due to either increased degradation or reduced synthesis of RAB35 protein upon LRRK2 inhibitor treatment. The role of LRRK2-mediated RAB35 phosphorylation in the reduction of RAB35 levels would be another interesting topic for investigation. To address these problems, one has to overcome the technical difficulties of measuring RAB35 phosphorylation in vivo.

In conclusion, LRRK2 kinase phosphorylates RAB35 and causes dyshomeostasis of α-synuclein aggregate trafficking through the endosomal pathway (Supplementary Fig. 12). Consequently, propagation of α-synuclein is augmented by LRRK2-mediated RAB35 phosphorylation. Thus, LRRK2 kinase activity and RAB35 function might be therapeutic targets for slowing or halting disease progression.

## Methods

**Materials**. The following antibodies were used in this study: α-synuclein monoclonal antibody (1:1,500 dilution, #610787, BD Biosciences, San Diego, CA), α-synuclein monoclonal antibody #274 (1:1,500 dilution), mouse anti-human α-synuclein clone syn211 (1:10,000 dilution, #36-008, Merck Millipore, Darmstadt, Germany), β-actin monoclonal antibody AC-15 (1:10,000 dilution, A5441; Sigma-Aldrich, St. Louis, MO), Anti-Myc polyclonal antibody (1:10,000 dilution, ab9106, Abcam, Cambridge, MA), Anti-LRRK2 (phospho S910) antibodies, UDD1 15(3) (1:1,000 dilution, ab133449, Abcam), Anti-LRRK2 (phospho S935) antibodies, UDD2 10(12) (1:1,000 dilution, ab133450, Abcam), Anti-LRRK2 (phospho S955) antibodies, MJF-R11 (75-1) (1:1,000 dilution, ab169521, Abcam), RAB1 monoclonal antibody (1:100 dilution, sc-515308, Santa Cruz Biotechnology, Santa Cruz, CA), RAB8 monoclonal antibody (1:500 dilution, #610844, BD Biosciences), RAB35 polyclonal antibody (1:100 dilution, #11329-2-AP, ProteinTech Group Inc, Chicago, IL), 3F10 rat monoclonal antibody (#12158167001, Sigma-Aldrich), HRP-conjugated goat anti-mouse IgG (H+L) (1:3,000 dilution, #172-1011; Bio-Rad Laboratories, Hercules, CA), and HRP-conjugated goat anti-rabbit IgG (H+L) (1:3,000 dilution, #170-6515 Bio-Rad Laboratories). Fluorescence dye-conjugated goat anti-rabbit IgG was purchased from Jackson Immunoresearch Laboratories (West Grove, PA). TO-PRO-3 iodide (T3605) and Q tracker 585 cell labeling kit were purchased from Invitrogen (Carlsbad, CA).

**Strains and culturing of nematodes**. All worms were maintained on nematode growth medium (NGM) plates seeded with *Escherischia coli* (*E. coli*) strain OP50 and grown according to standard procedures at 20 °C[67]. Wild-type Bristol N2 was obtained from the *Caenorhabditis* Genetics Center (CGC; University of Minnesota, St. Paul, MN, USA). The mutant lines *lrk-1(tm1898)* and *rab-35(tm2058)* were acquired by *C. elegans* National BioResource Project (NBRP; Tokyo Women's Medical University School of Medicine, Tokyo, Japan).

**Plasmid construction for *C. elegans***. Plasmids for *C. elegans*, including P*myo-2*::V1S and P*flp-21*::SV2-ICR-DsRed vector, were generated as described below.
The *lrk-1* promoter (P*lrk-1*) was amplified from genomic DNA obtained from wild-type N2 worms. P*lrk-1*::V1S was generated by replacing the PCR product with the P*myo-2*::V1S vector[8].
To make vectors containing human RAB35 (*hRAB35*) wild-type (WT) and mutants, a sense primer containing a *Sal*I site, 5′- ATC GTC GAC ATG GCC CGG GAC TAC GAC CA-3′ and an antisense primer containing *Bgl*II site, 5′- TAG AGA TCT CTA GCA GCA GCG TTT CTT TCG-3′ were used to amplify the *hRAB35* WT obtained from pcDNA3.1 Myc-His *RAB35* vector, generous gifts from

BD Lee (Kyung Hee University, Seoul, Korea). The V1S fragment of P*lrk-1*::V1S was replaced by the PCR-amplified *hRAB35* WT fragment to construct P*lrk-1*::*hRAB35* WT. Based on P*lrk-1*::*hRAB35* WT, P*lrk-1*::*hRAB35* Q67L, which was created by substitution of a leucine for a glutamine amino acid at the 67th codon site, and P*lrk-1*::*hRAB35* T72A, which was produced by replacing a threonine with an alanine amino acid at the 72th codon position, plasmids were generated using a QuikChange Site-Directed Mutagenesis Kit (#200521, Stratagene, La Jolla, CA).

**Generation of BiFC transgenic lines**. To analyze the effects of *lrk-1*, *rab-35* on cell-to-cell α-synuclein propagation, the mutant lines, *lrk-1(tm1898)* and *rab-35 (tm2058)*, were used. P*myo-2*::V1S and P*flp-21*::SV2-ICR-DsRed plasmids were co-injected into the gonads of late L4-stage mutant worms with a selection marker, pRF4 which expresses a mutant collagen gene, *rol-6(su1006)*[68], to generate transgenic line expressing the BiFC pair with α-synuclein . For the analysis of the effects of phosphorylation in RAB35 on α-synuclein propagation, P*myo-2*::V1S, P*flp-21*::SV2-ICR-DsRed and P*lrk-1*::*hRAB35* WT or P*lrk-1*::*hRAB35* T72A plasmids were co-injected into late L4-stage *rab-35(tm2058)* mutants with pRF4. In addition, P*myo-2*:: V1S, P*flp-21*::SV2-ICR-DsRed and P*lrk-1*::*hRAB35* Q67L were co-injected into late L4-stage *lrk-1(tm1898)* mutants with pRF4 to identify the relationship between LRK-1 and RAB35. After several transgenic lines containing the introduced plasmids were acquired, three representative BiFC transgenic lines in each mutant background were selected for experiments.

**Single-worm polymerase chain reaction (PCR)**. The genomic DNA obtained from a gravid single worm of each line was mixed with Ex Taq™ polymerase (RR001A, Takara Biotechnology, Kyoto, Japan), and subsequently PCR reaction for target genes was performed in Bio-Rad MyCycler PCR Thermal Cycler system (Bio-Rad Laboratories Inc., Hercules, CA)[8].

**Western blotting of *C. elegans***. Protein samples derived from each BiFC transgenic line were loaded onto 12% SDS-PAGE gels, and then transferred to a nitrocellulose membrane[8]. The following antibodies were used for immunoblotting in this study; monoclonal anti-α-synuclein antibody, Syn-1 (1:1,500 dilution; #610787, BD BioScience), polyclonal anti-RAB35 antibody (1:200 dilution; #11329-2-AP, ProteinTech Group Inc.,), HRP-conjugated goat anti-mouse IgG (H+L) (1:3,000 dilution, #170-6516, Bio-Rad Laboratories,), and HRP-conjugated goat anti-rabbit IgG (H+L) (1:3,000 dilution, #170-6515, Bio-Rad Laboratories). Chemiluminescence detection was performed with ECL prime solution (GE Healthcare Life Sciences Amersham, RPN2232) using the Amersham imager 600 (GE Healthcare Life Sciences, Marlborough, MA, USA).Images were quantified with Multi Gauge (v3.0) software (Fujifilm, Tokyo, Japan).

**Fluorescence microscopy of live worms**. Worms were collected, washed with M9 buffer (22 mM KH$_2$PO$_4$, 22 mM Na$_2$HPO$_4$, 85 mM NaCl, 1 mM MgSO$_4$), and then immobilized with M9 buffer containing sodium azide (S2002, Sigma-Aldrich). The sample was dropped on microscope cover glass (HSU-0101242, Marienfeld Laboratory Glassware, Lauda-Königshofen, Germany), and covered with a coverslip. All images of BiFC transgenic models were obtained using Olympus FV1000 confocal laser scanning microscopy (Olympus, Tokyo, Japan). The images were captured by using standard filters and appropriate lasers. Acquisition parameters were fixed for excitation intensity, gain value, and voltages on each imaging channel. For analysis of fluorescence intensity in pharynx, pharyngeal areas were selected by using drawing/selection tools in the FV10-ASW 3.1 viewer (Olympus, Tokyo, Japan). Subsequently, the integrated fluorescence intensity and area of the selected regions of interest (ROIs) within the pharynx were measured using FV10-ASW 3.1 analysis software (Olympus, Tokyo, Japan). The BiFC fluorescence intensity was determined as the relative fluorescence intensity per area (μm$^2$) in pharynx region. The values represent the average intensities of several worms in each line.

**Pharyngeal pumping analysis**. Pharyngeal pumping of worms, including wild-type N2, mutants and transgenic lines, was counted for 1 min at room temperature (RT) using Axio Observer A1 inverted microscope (Carl Zeiss MicroImaging Inc., Göttingen, Germany). The values were expressed as Pumps Per Minute (PPM).

**Life span assay**. L4 -stage worms synchronized from eggs were placed on NGM plates containing 100 μM 5-fluoro-2′-deoxyuridine (FUdR; F0503, Sigma-Aldrich) to prevent progeny production. The number of worms that were alive or dead was monitored and scored at regular intervals of 1 or 2 days. Worms that were ruptured, burrowed, or crawled off the plates were censored but included in the life-span analysis as censored animals. The mean life span and life span assays were analyzed using online application for the survival analysis (OASIS; http://sbi.postech.ac.kr/oasis/surv/)[69].

**Vagal AAV injections in rats**. Recombinant AAV (serotype 2 genome and serotype 6 capsid) was used for transgene expression of human wild-type α-synuclein under the control of the human Synapsin1 promoter (Sirion Biotech, Martinsried, Germany). The AAV genome contained a woodchuck hepatitis virus post-

transcriptional regulatory element and a polyadenylation signal sequence downstream to the human α-synuclein sequence. All experiments were approved by the ethical committee of the State Agency for Nature, Environment and Consumer Protection in North Rhine Westphalia. Surgical procedures were performed as described below. Young adult (200–250 g) female WT controls and *Lrrk2*−/− rats on the Charles River Long-Evans background strain (LEH-*Lrrk2* tm1sage, Horizon Discovery, Saint Louis, MO, USA) were anesthetized with 2% isoflurane mixed with $O_2$ and $N_2O$. An incision was made at the midline of the rat neck, the left vagus nerve was isolated, and the vector solution (2 μl) was injected at a titer of $7.5 \times 10^{12}$ genome copies ml−1. WT and *Lrrk2*−/− rats were treated in parallel. At 8 and 12 weeks after AAV injections, animals were killed with pentobarbital and perfused through the ascending aorta with saline and 4% (w/v) paraformaldehyde. Brains were immersion-fixed in 4% paraformaldehyde and cryopreserved in 25% (w/v) sucrose solution[24].

**Molecular biology analyses in *Lrrk2*−/− rats**. To assess expression of LRRK2 in WT and *Lrrk2*−/− rats, total RNA was extracted from paraformaldehyde-fixed cortical tissue using the RecoverAll™ Total Nucleic Acid Isolation kit (Ambion, TX, USA), and cDNA was synthesized using SuperScript™ VILO™ Master Mix (Thermo Fisher, Carlsbad, CA, USA). The following primers were used for RT-PCR analysis: 5′-CAC TTC ATG ACC CAG AGA GCC CTG-3′ forward and 5′-TGT GCC CAC CAG AAT CAC CG-3′ reverse (*Lrrk2*); 5′-GAC CGG TTC TGT CAT GTC G-3′ forward and 5′-ACC TGG TTC ATC ATC ACT AAT CAC-3′ reverse (*Hprt*). PCR products were separated on agarose gel.

**Histological analyses in the AAV rat model**. Coronal brain sections (40 μm) were cut using a freezing microtome, and free-floating sections were processed for immunohistochemistry[24]. Mouse anti-human α-synuclein clone syn211 (1:10,000, Merck Millipore, Darmstadt, Germany) and rabbit anti-LRRK2 clone c41-2 (1:250, Abcam, Cambridge, MA) were used as primary antibodies prior to incubation in biotinylated-secondary antibody solution (Vector Laboratories, Burlingame, CA, USA). Color reaction was developed using 3,3′-diaminobenzidine kit (Vector Laboratories). Brightfield images of syn211-immunoreactive axons were generated from stacks collected at 0.28 μm intervals using a Zeiss Observer.Z1 Microscope (Carl Zeiss) equipped with an AxioCam IC camera and a 100× Plan-Apochromat (1.40) objective. Stacks were converted into Z-projections using Extended Focus module (ZEN 2 software, Carl Zeiss). All histological quantifications were carried out by investigators blinded to experimental group. For stereological cell counts of neurons in the dorsal motor nucleus of the vagus nerve, medulla oblongata sections encompassing the entire nucleus were stained with syn211 and counterstained with cresyl violet[25]. Length and density of syn211-immunoreactive axons were measured on three serial sections of the left (injected side) pons (Bregma: -9.70 to -9.22 mm) where an area encompassing the coeruleus/subcoeruleus complex and the parabrachial nuclei was delineated. Measurements were performed using the Space Balls stereological probe (Stereo Investigator version 9, MBF Biosciences)[25]. The number of syn211-positive axons was counted in the left (injected side) pons, midbrain and forebrain using an Axioscope microscope (Carl Zeiss) under a 20× Plan-Apo objective. Sections were collected at pre-defined Bregma coordinates: −9.48 (pons), −7.80 mm (caudal midbrain), −6.00 mm (rostral midbrain) and −2.40 mm (forebrain).

**Cell culture and α-synuclein expression**. Human neuroblastoma SH-SY5Y cells (CRL-2266, ATCC, Manassas, VA) were cultured and differentiated[26]. Briefly, cells were subcultured every 2 days at 37ºC in humidified air with 5% $CO_2$ in Dulbecco's modified eagle's medium (DMEM) (SH30243.01, HyClone, Logan, UT) containing 10% fetal bovine serum (SH30396.03, HyClone), 100 units mL−1 penicillin, and 100 units mL−1 streptomycin (#15140-122, Gibco, Grand Island, NY). For differentiation, cells were incubated with growth media containing 50 μM all-*trans*-retinoic acid (R2625, Sigma-Aldrich). In the case of overexpression of human α-synuclein, differentiated SH-SY5Y cells were infected with recombinant adenoviral vector (serotype Ad5, CMV promoter) containing human α-synuclein cDNA at a multiplicity of infection of 33.3. For overexpression of human LRRK2, SV2 cells were infected with a recombinant adenoviral vector (serotype Ad5, CMV promoter) containing either human *LRRK2* WT, *LRRK2* G2019S, or *LRRK2* D1994A cDNA at a multiplicity of infection of 2. For transduction of RAB proteins, SV2 cells were transduced by lentiviral vectors for 6 h. After washing out the viruses, cells were incubated with normal growth media. In the case of BiFC co-culture, V1S and SV2 cells were mixed on a coverslip and cultured for 3 days.

**Preparation of conditioned medium**. Differentiated SH-SY5Y cells were infected with recombinant adenoviral vector containing human α-synuclein cDNA as described above. After 2 days of infection, cells were washed three times with fresh DMEM and incubated with serum-free DMEM for 18 h. Medium was collected and centrifuged at 250 g for 10 min at 4 ºC to remove cell debris.

**Accumulation of internalized α-synuclein aggregates**. Fully differentiated cells were infected with either Mock, LRRK2 WT, LRRK2 G2019S mutant, or LRRK2 D1994A mutant as described above. After 3 days of incubation, cells were treated with 5X concentrated conditioned media obtained from SH-SY5Y cells

overexpressing human α-synuclein. After incubation at 37 °C in a $CO_2$ incubator for the indicated times, the levels of cumulative α-synuclein were measured by western blotting. The levels of internalized α-synuclein were normalized with those of β-actin. The relative levels of α-synuclein in steady states represent the ratio to the maximum level of the internalized total α-synuclein in mock-transduced SH-SY5Y cells.

**Secondary secretion of BiFC-positive α-synuclein**. The secretion of Venus BiFC-positive α-synuclein aggregates was performed as described below. After 2 days co-culture of V1S cells and SV2 cells, the culture media was collected. Following the centrifugation to remove the cell debris, the secondary secretion of BiFC (+) α-synuclein aggregates was analyzed by measuring the Venus BiFC fluorescence in the culture media obtained from co-culture[26].

**Inhibition of LRRK2 kinase activity in cell culture**. Inhibition of LRRK2 kinase activity was accomplished by treatment with 5 μM of LRRK2 kinase inhibitor, HG-10-102-01 (#438195, Calbiochem, La Jolla, CA) for 2 days. The inhibition of LRRK2 kinase activity was confirmed by the reduction in the autophosphorylation of LRRK2.

**SH-SY5Y co-culture assay**. The recipient cells were labeled with a Qtracker 585 cell labeling kit[70]. Q dot labeled recipient cells were co-cultured for 3 days with donor SH-SY5Y cells infected with α-synuclein adenoviral vector. After immunofluorescence staining, the transmitted α-synuclein was observed by using an Olympus FV1000 confocal laser scanning microscope (Olympus, Tokyo).

**Immunofluorescence staining**. Cells grown on poly-L-Lysine-coated coverslips were fixed with 4% paraformaldehyde in Phosphate-buffered saline (PBS). After permeabilization with 0.1% Triton X-100 in PBS, cells were incubated with blocking solution (5% bovine serum albumin/3% goat serum in PBS). After blocking, cells were incubated with primary antibodies diluted in blocking solution and washed three times with ice-cold PBS. After incubation with fluorescent dye-conjugated secondary antibodies diluted in blocking solution, nuclei were stained with TOPRO-3 iodide (T3650, Invitrogen). Cells were mounted onto slide glasses in the presence of Prolong Gold Antifade Reagent (P36930, Invitrogen). Images were obtained by using Olympus FV1000 confocal laser scanning microscope[70].

**Colocalization assay**. For the colocalization assay in cell culture models, 300 cells which contain transmitted α-synuclein were analyzed from each treatment per experiment. Colocalization was analyzed by using NIH Image J, Fiji software. Pearson's coefficients analysis was performed using the Coloc 2 plugin in ImageJ Fiji. The values were obtained from three independent experiments.

**Preparation of cell extracts**. Cells were washed with ice-cold PBS and lysed in extraction buffer (1% Triton X-100, 1% (v/v) protease inhibitor cocktail (P8340, Sigma-Aldrich) in PBS. Cell lysates were incubated on ice for 10 min and centrifuged at 16,000×g for 10 min After collecting supernatant, Triton X-100 soluble fraction, Triton X-100 insoluble fraction was resuspended with 1X Laemmli sample buffer and sonicated briefly.

**Myc-6x histidine tagged LRRK2 pull-down**. For pull-down of myc-6x histidine tagged LRRK2, after overexpression, cell lysates were loaded to equilibrated Ni-NTA agarose (Qiagen, Valencia, CA). After incubation for overnight at 4ºC, beads were washed with washing buffer A (1% Triton X-100 and 20 mM imidazole in PBS), followed by washing twice with buffer B (1% Triton X-100 and 30 mM imidazole in PBS) and buffer C (1% Triton X-100 and 50 mM imidazole in PBS). The protein conjugates were eluted with 2x Laemmli and boiled at 95 °C for 10 min Eluates were analyzed by western blotting. The relative levels of eluted RAB35 were calculated the ratio versus eluted RAB35 level in SH-SY5Y cells overexpressing LRRK2 WT.

**Immunoprecipitation of HA-tagged human RAB35**. In the case of immunoprecipitation of HA-tagged human RAB35, the cell lysates were incubated with either control rat IgG or 3F10 antibody (Sigma-Aldrich), a rat monoclonal antibody against HA tag for overnight at 4 °C. Samples were incubated with protein A/G agarose beads (Thermo-Fischer Scientific, Waltham, MA) for 2 h at 4 °C. After washing with RIPA buffer, the protein was eluted with with 2x Laemmli and boiled at 95 °C for 10 min Eluates were analyzed by western blotting. The relative levels of LRRK2 in eluate fraction were calculated the ratio versus the amount of eluted LRRK2 in SH-SY5Y cells overexpressing LRRK2 WT.

**Western blotting**. Protein samples derived from cells were loaded onto 8% or 12% SDS-PAGE gels, and then transferred to a nitrocellulose membrane. After blocking with skim milk, membranes were incubated with indicated primary antibodies. The membranes were incubated with HRP-conjugated secondary antibodies and reacted with Amersham ECL prime western blotting substrate (GE healthcare). Images were obtained using the Amersham imager 600 (GE Healthcare Life Sciences,

Marlborough, MA, USA)and analyzed with Multi Gauge (v3.0) software (Fujifilm, Tokyo, Japan)[26].

**Animal treatment**. Transgenic mice expressing wild-type human α-synuclein under the mThy1 promoter were used[71]. Briefly, age-matched (9-month-old) 12 non-tg and 12 α-syn-tg mice were injected intraperitoneally with either vehicle (6 mice per group, 10% DMSO, 45% PEG400, 45% Water) or LRRK2 inhibitor (6 mice per group, 10 mg kg$^{-1}$ HG-10-102-01) for 4 weeks (injected 5 days per week). The right hemi-brains were snap-frozen and stored at −70 °C for biochemical analysis, while the other hemi-brains were fixed with 4% paraformaldehyde for neuro-pathological analysis. All animal procedures were approved by the UCSD Institutional Animal Care and Use Committee under protocol #S02221.

**Immunohistochemical analysis**. Blind-coded sagittal brain sections were treated with primary antibodies against α-synuclein (Syn-1, BD Biosciences) or phosphorylated- α-synuclein (pSer129, Wako) at 4 °C for overnight. To detect proteinase K (PK) resistant α-synuclein, brain sections were pre-incubated with PK (10 µg ml$^{-1}$) for 8 min, and immunostained with antibody against total α-synuclein (BD Bioscience). Next day, the sections were incubated with biotinylated-secondary antibodies and detected with avidin D-HRP (ABC elite, Vector Laboratories, Burlingame, CA). To determine α-synuclein neuropathology, the brain sections were imaged by Olympus BX41 microscope. The levels of immunoreactivity were determined by optical density analysis using Image Quant 1.43 program (NIH). The numbers of inclusion positive cells were determined per field (100 square µm) of each animal based on cell body recognition using Image Quant 1.43 program (NIH)[72].

**Double immunofluorescence labeling analysis**. Blind-coded brain sections were immunolabelled with indicated primary antibodies to determine the co-localization between α-synuclein/RAB35, and α-synuclein/cathepsin D. Immunoreactivities were detected with either Tyramide Signal Amplification Direct system (PerkinElmer) or FITC-tagged secondary antibodies. Sections were imaged with a Zeiss 63X (numerical aperture 1.4) objective on an Axiovert 35 microscope (Zeiss) with an attached MRC 1024 LSCM system (Bio-Rad). The levels of co-localizations were analyzed in 10 random chosen fields from double-labeled sections using Image J program (NIH)[72].

**Brain tissue extract and western blot analysis**. Brains of mice were homogenized with lysis buffer (1 mM HEPES, 5 mM benzamidine, 2 mM 2-mercaptoethanol, 3 mM EDTA, 0.5 mM MgSO$_4$, 0.05% NaN$_3$, protease inhibitor cocktail III, phosphatase inhibitor cocktail II), and centrifuged at 100,000×$g$ for 1 h to obtain cytosolic and particulate (membrane bound) fractions. The fractions were loaded onto 4–12% Bis-Tris SDS-PAGE gels (Invitrogen), transferred onto PVDF membranes, and blocked with bovine serum albumin. After an incubation with indicated primary antibodies, the membranes were incubated with HRP-conjugated secondary antibodies (American Qualex, San Clemente, CA), and reacted with ECL western blotting substrate (PerkinElmer, Waltham, MA). Chemiluminescence detection and densitometry analysis were performed using Versadoc XL imaging apparatus and Quantity One (Bio-rad, Hercules, CA).

**Statistical analysis**. Values shown in the figures were presented as mean ± S.E.M. To determine the statistical significance, P values were calculated by means of unpaired, two-tailed Student's $t$-tests, one-way ANOVA with Tukey's post hoc tests, one-way ANOVA with Dunnet's post hoc test, two-way ANOVA with Dunnet's post hoc test, or two-way ANOVA with Tukey's post hoc test using GraphPad Prism version 7.00 for Mac, Graphpad Software, La Jolla California USA, (www.graphpad.com).

**Data availability**. All data and/or analyses generated during the current study are available from the corresponding author upon request

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

## Acknowledgements

This work was supported by the National Research Foundation (NRF) grant funded by the Korean Government (MEST) (NRF-2015R1A2A1A10052540 to S.J.L. and NRF-2016R1C1B2013940 to E.J.B.). This work was also supported by grant funded by Seoul National University Hospital and the Paul Foundation.

## Author contributions

E.J.B., C.L., and H.J.L. designed and performed cell biological experiments and analyzed the data. D.K. designed and performed *C. elegans* experiments and analyzed the data. C. K., M.M., A.A., and E.R. designed and performed mouse experiments and analyzed the data. E.M. designed the mouse experiments, analyzed the data, and wrote sections of the paper. A.U. performed surgeries and carried out histological analyses and neuronal counts in AAV-injected rats; M.K. performed molecular biology analyses and axonal quantification in rats; D.A.D.M. designed experiments in rats, analyzed data and wrote sections of the paper. G.R.J., J.R.B., and B.D.L. provided the plasmids for RAB35 and essential information on RAB35 phosphorylation. S.J.L. conceived and oversaw the study, designed experiments, analyzed the data, and wrote most of the paper with E.J.B.

## Additional information

**Competing interests:** S.J.L. received research grants from and is a stock-holder of Abl Bio. The remaining authors declare no competing interests.

