## [Peer Review File · Nature Communications]

Reviewers' comments:

Reviewer #1 (Remarks to the Author):

The manuscript "LRRK2 kinase regulates intercellular transmission of alpha-synuclein aggregates through phosphorylation of Rab35" uses a variety of model systems to show that LRRK2 control over a-synuclein uptake and propagation are dependent on interactions with Rab35. The strengths of the manuscript are numerous and include the use of model systems that increase in complexity to test the Rab35-LRRK2 hypothesis, from the cell lines to *c. elegans* to two different but complementary models in rodents. Further, the data are generally very convincing in recording large and clear-cut effects for most of the interactions, suggesting the conclusions may be quite robust. The impact is high in the initial formation of a pathway that has thus far remained elusive: the molecular mechanism of interactions between LRRK2 and a-synuclein aggregation.

There are some minor experimental omissions that should be considered, as well as some organization details with respect to the text:

Minor text revisions:

1- The term 'intercellular transmission', for example in the title and in the text, is verbose compared to the term commonly used in the field- 'propagation'. Further, the word 'transmission' may not be very accurate, because the aggregates themselves may not 'transmit' as it may be assumed in this work (yet not demonstrated). The authors and many others can clearly show a-synuclein inclusion propagation through the brain in rats and mice and in vitro. It is advised to revise the relevant portions of text.

2- Elements of the introduction and discussion sections are switched. The introduction rambles on and does not introduce the background literature relevant to the questions. Instead, a loose discussion of how pathology may correlate with clinical features in PD and LRRK2-PD is presented. It is advised that the revised introduction be centered on the relevant literature which is largely references 22-24 that deal with the subject at hand. Further, if reference 4 is to be discussed (which it probably should not be), then it would be prudent to also give fair weight to counter-reviews that oppose those interpretations published recently by Halliday, Surmier, Brundin and others. These could be saved for the discussion or omitted.

Experimental considerations:

1- The LRRK2 Rab35 phosphorylation data (included 'only for reviewers') needs to be presented in this paper. It is unacceptable to be saved for a different publication, for example, and not be included as a supplemental figure, given the focus on Rab35 in this paper.

2- In figure 2a, lysates from cells not treated the synuclein-positive media needs to be shown as western blots. These blots may be blank. Also, the antibody should be given in the figure legends (assuming this is the synuclein antibody from the Methods?, but it could also be antibodies to venus..)

3- In interpretation of Figure 2, that data could all be explained by enhanced uptake of synuclein mediated by LRRK2. Or, it could be both increased uptake as well as increased release of a-synuclein. The later was not tested but needs to be tested for interpretation. It could be that LRRK2 upregulates both uptake AND release, this needs to be determined in this model.

3- In Figure 3, the co-localization analysis is of very low value. The reason is that the Rab35 expression is mostly diffuse, likely due to over-expressing the protein for too long in the cell lines,

many groups express for only a few hours to preserve a proper ratio of membrane-to-cytosolic ratio. Further, the interpretation of α -synuclein 'aggregates' may simply be α -synuclein dimers localized to endosomes. The IPs are much stronger evidence. If the Author's want to keep the analysis, the co-loc should be re-performed under conditions that identify α -synuclein aggregates (e.g., thioflavin), and preserve Rab35 membrane localization, and present convincing photomicrographs (super-resolution may be needed for convincing co-loc).

Reviewer #2 (Remarks to the Author):

The article by Bae et al "LRRK2 kinase regulates intercellular transmission of α -synuclein 2 aggregates through phosphorylation of Rab35." addresses an issue of great interest to the PD field and more generally to those interested in cell biology. The experiments and the document are generally well composed, and mostly appropriate. I find the use of the array of techniques good, but somewhat disjointed. Although there are many strengths to the work, which I feel should be championed, I feel the writing is massively over-simplified and the conclusions far too broad based on the evidence presented. I am enthusiastic for a lot of the work, and recommend that the editors **STRONGLY** consider a resubmission, but in its' present state the document is crushed by brevity, over interpretation of the data, and the perceived need to 'tick the boxes' of a novelty-driven, high impact journal. The impact of the work wouldn't be diminished by a more honest interpretation, but might be more appreciated by the audience.

For example (this would be my major concern), throughout the document there is little self-awareness or self-criticism, rather the furthest possible 'important' extrapolation of the data is presented as the simplest interpretation of the results.

An overarching theme for the way the document is written is that Lrk-1 is not just a homologue, but is analogous to LRRK2.

C. elegans (Lrk-1) and *Drosophila* (Lrrk) genes have an ancient origin distinct from that of LRRK2, while perhaps a conserved homolog for mammalian LRRK1 (important during embryogenesis and development in mammals, but absent very soon after birth) there is little evidence of a good relationship between Lrk-1 and LRRK2 which is expressed exclusively (without LRRK1) in adult mammalian tissue. This is not species snobbery from this reviewer, it is more the lack of transparency and overly hyperbolic writing in the document that stimulates such a comment. A more unbiased writing style would improve the reviewers appreciation of the data, rather than the drum-banging.

Similarly, much is made of the Lrk-1 hypomorphs/functional knockouts being evidence for LRRK2's role in transmission of synuclein; however, in other reports the worm Lrk-1 mutants have several other alterations (odd axonal arborizations, presumably also synaptic function, axonal proteins localized in dendrites etc. e.g., Sakaguchi-Nakashima 2007) which might impact upon synuclein transmission between cells independently of whatever (the absent) LRRK2 protein might do. Many of these alterations might be due to disturbed neuronal development or general worm health in the absence of a LRRK1/2 homologue, and thus the data should not be used to support an argument based on impairment of 'LRRK2' function that would normally facilitate synuclein transmission between cells (at least without open discussion of other possibilities).

This reviewer doesn't even see these shortfalls as a problem for the current data set, merely the language used in the description of the data.

Another overall concern, not addressed, is that much of the *elegans* data is assayed at day 13... when ~20% Lrk-1 and 50% of asyn worms are already dead... are animals dying early because inclusions and synuclein propagation (as inferred) or prior to / in absence of these phenomenon

Specific concerns:

- 1) The introduction makes mention of the fact that most LRRK2 PD cases present with synucleinopathy, yet should also mention that nearly half do not.
- 2) It is inappropriate to state that BiFC transgenic *C. elegans* lines are LRRK2-deficient (*lrk-1*). Besides the arguments listed above, at least one report states LRRK2 inhibitors don't work on *lrk-1* (Yao 2013, *Hum Mol Genet*).
- 3) Authors state "These results indicate that LRRK2 plays an important role in α -synuclein transmission in *C. elegans*." besides from LRRK2 not being present in *C. elegans*, *lrk-1* deficient worms not otherwise normal... maturation rates, feeding, movement, among other alterations could account for a reduction in synuclein transmission which is entirely indirect
- 4) Even figure legends and data are labelled LRRK2 inappropriately.
- 5) Fig 1 J. Is this drawn from a single example animal / section or is this aggregated imagery composed from several animals? Fig. 1K., number of projections is an odd metric. From the images it isn't as if there is a percentage (others labelled), or knowledge of whether signal is multiple parts of one axon... this might be better quantified as signal area / densitometry analyses?
- 6) Figure 2. For cell lines infected with LRRK2 it is important to show the levels of overexpression of each construct, also the number of cells and their proximity to each other must be approximately equal to make sense of 'transmission' likelihood?
- 7) why is there no regular sized synuclein in the blot? Even in insoluble fractions, should be... unless there is none in the cell line? Can the authors also provide soluble? Or are cells internalising aggregated synuclein? It is also unclear from the text what synuclein is "relative" to - presumably actin control... would be better assessed relative to total synuclein in the cell?
- 8) Problems with data presentation: number of cells expressing seems inappropriate... all are 1 yet with variability? Most GS must contain 1.4 with 10% variance with odd cell being way off? Also, n is not 3 surely (as stated)?, n is likely 200 from 3 independent experiments. This needs to be much clearer throughout; change the writing or the presentation (scatter plots with overlaid mean \pm SEM would help)
- 9) No explanation of blue signal DAPI? Again to interpret such data we need to know TF efficiency of each, how many cells, how close they are to each other.
- 10) Need to address why inhibitor only works in OE not mock. Need to check levels of OE, density in the cells, proximity etc? Residual effects not altered by inhibitor... no endogenous LRRK2? ALSO mutant not different from BAC GS - this is not as expected if a kinase-dependent phenomenon, expect expression levels may be different.
- 11) Fig 3. Colocalisation experiments must be by coefficients (eg pearsons R) puncta analyses and overlap / coloc number are fine, but require extra information of puncta number, size and intensity to make sense of number colocalised. Similarly n is not clearly reported again (see point 8)
- 12) Does rab35 LRRK2 interaction occurs with non-overexpressed endogenous proteins also? Seems example blots are missing in Fig3 D? Also a casual glance makes it look like MORE rab35 associated with LRRK2 in WT than GS in c & d (unclear in 3F10 example). This should be quantified as might be driving force for alterations, or at very least addressed in text
- 13) other than n's being hard to decipher... statistical analyses inappropriate; presented for 2way

RM ANOVA yet paired ttest???

14) Figure 4 n again hard to work out. Life span here and other two in WT shows they seem to mostly ALL die on the same day - here is day 16, p day 14, fig5h is 14. Other than Fig 4h is hard to know how much a difference (1day) is due to variability or treatments... scatters and join bars between replicates would help, e.g., in 3 experiments is WT always the same sine different? Is randomness in the 200?

15) Figure 6 labelled "transmission". How is this clearly transmission, and not reduced expression in certain cell types? Which cells are expressing, are they assaying in areas which don't express on the transgene?

16) Figure 6 Puncta analyses... again if percent coloc. need details on number and intensity of both signals. Better with Pearson's. That said rab increases in intensity are interesting so need to show more e.g., puncta size, number etc. rab levels by wb etc.

17) Figure 6. What is the evidence for target engagement in these experiments (not others) e.g., LRRK2 phosphorylation. Also again the lack of detail on statistical analyses are alarming, was a stats form filled in? Where are details of the ANOVA F-values and df, residuals etc throughout?

18) Supplemental fig 6. The inhibitor reduces LRRK2 p but also seemingly Rab35 and aSyn levels. While they show the rescue attempt is superseded by Rab constitutively active, these conclusions require evidence for whether LRRK2 protection is mimic by reduction in pRab?

Reviewer #3 (Remarks to the Author):

The manuscript by Bae and colleagues presents data that "LRRK2 regulates the intercellular transmission of α -synuclein aggregates via phosphorylation of Rab35." The study explores intercellular transmission of the Parkinson's disease-associated protein α -synuclein using C.elegans, human SH-SY5Y cells, rats and α -synuclein transgenic mice. The study offers interesting data that LRRK2 plays a role in regulating α -synuclein transmission via the phosphorylation of Rab5. On the other hand, the manuscript is confusing in many places because of the switching between the phrases " α -synuclein transmission" and " α -synuclein aggregate transmission." The main claim—that α -synuclein aggregates are transmitted from cell-to-cell—is not well-supported by the data. Some specific points to address are as follows.

1) lines 78-104:

" α -synuclein transmission" is used on lines 83-4, 94, 99, and then the paragraph ends by switching to the phrase " α -synuclein aggregate transmission" on line 103.

Why the switch from " α -synuclein transmission" to " α -synuclein aggregate transmission"? Between lines 83-102 there is no data that shows that α -synuclein aggregates were transmitted. How was it ruled out that α -synuclein monomers exchanged between the nerve and muscle cells in wild type worms? Perhaps in the *lrrk-1* worms the mutation inhibits release of the monomer. More on this point, in Figure 1a, the right-hand images of the donor and recipient cells shows that both cells release monomers, which are not aggregates.

2) Figure 2, legend: "(c-e) Alterations in LRRK2 kinase activity regulates cell-to-cell transmission of α -synuclein. V1S cells were co-cultured with either myc conjugated WT, GS, or DA LRRK2 infected SV2 cells. The transmission of α -synuclein aggregates were analyzed by measuring BiFC (+) cell (%)."

a) How do you know that α -synuclein aggregates were transferred between donor and recipient cells? Were α -synuclein aggregates detected in the culture media? Were monomers detected in the

culture media?

b) What does "co-aggregates" mean?

3) Disconnect between text (lines 55-174) regarding Figure 3 and the legend to Figure 3.

" α -synuclein transmission" is used four times between lines 55-174. Then a shift occurs in the legend to " α -synuclein aggregates": "(e, f) V1S cells were co-cultured with LRRK2 overexpressing SV2 cells which were co-infected with either mock or Rab35 DN. After 3 days of co-culture, transmission and co-aggregation of α -synuclein aggregates were calculated by measuring the BiFC (+) cell (%).

This jarring disconnect between the main text and the figure legend. There is no evidence that aggregates were transmitted. Just because BiFC aggregates were detected does not necessarily mean that aggregates were transmitted. How can you rule out that some cells release α -synuclein monomers, which then are endocytosed by neighboring cells, and the increase in the intracellular concentration of monomers causes aggregation inside the recipient cells? Were BiFC aggregates detected in the cell culture medium? Monomers?

4) Figure 1 legend, (b) BiFC fluorescence in wild type and *lrrk-1(tm1898)* mutant worms at day 13. The red arrows: BiFC signals, the red arrowheads: inclusions in pharynx.

There are no red arrows in the figure; only red arrowheads.

5) Figure 1 legend, (c) Quantification of BiFC fluorescence at day 13. Twenty worms for each line were used.

The BiFC fluorescence values in plot c. More information is needed as to how the values (4,000 and 2,000) were obtained. Are you averaging intensities over the entire green portions of the worms?

6) line 155 LRRK2-mediated Rab35 phosphorylation regulates α -synuclein transmission paragraph and figure (Fig. 3)

You can help the Reader if you state explicitly what cells you are using in this paragraph. For example, in line 167 it says "...in the recipient cells." It is clearer to say "in the SH-SY5Y recipient cells." Same for the legend to Figure 3. Nowhere in the legend does it say what cells are being used.

7) Figure 4 a, b. (a, b) BiFC fluorescence in wild-type and *rab-35* transgenic models at day 13. White arrows: BiFC signals, red arrowheads: inclusions in pharynx. Twenty worms for each line were used.

It is not clear how the quantitation was done. How are the average fluorescence values for "BiFC signals (white arrows)" obtained. In the *rab-35* worm (Fig. 4a) there are two white arrows, which point to structures that have very different fluorescence intensities. It looks like there is significant variation in the intensities of these structures. That variation does not seem to be reflected in the relatively small error bar on the *rab-35* black bar in Fig. 4b.

8) Unless better evidence can be obtained that α -synuclein aggregates were transferred intermolecularly, the title should be changed.

Reviewer #1 (Remarks to the Author):

*The manuscript "LRRK2 kinase regulates intercellular transmission of alpha-synuclein aggregates through phosphorylation of Rab35" uses a variety of model systems to show that LRRK2 control over a-synuclein uptake and propagation are dependent on interactions with Rab35. The strengths of the manuscript are numerous and include the use of model systems that increase in complexity to test the Rab35-LRRK2 hypothesis, from the cell lines to *C. elegans* to two different but complementary models in rodents. Further, the data are generally very convincing in recording large and clear-cut effects for most of the interactions, suggesting the conclusions may be quite robust. The impact is high in the initial formation of a pathway that has thus far remained elusive: the molecular mechanism of interactions between LRRK2 and a-synuclein aggregation.*

There are some minor experimental omissions that should be considered, as well as some organization details with respect to the text:

Minor text revisions:

1- The term 'intercellular transmission', for example in the title and in the text, is verbose compared to the term commonly used in the field- 'propagation'. Further, the word 'transmission' may not be very accurate, because the aggregates themselves may not 'transmit' as it may be assumed in this work (yet not demonstrated). The authors and many others can clearly show a-synuclein inclusion propagation through the brain in rats and mice and in vitro. It is advised to revise the relevant portions of text.

Response: As per reviewer's suggestion, the term 'intercellular transmission' has been changed to 'propagation' throughout the text. The title is now 'LRRK2 kinase regulates propagation of a-synuclein via phosphorylation of rab35.'

2- Elements of the introduction and discussion sections are switched. The introduction rambles on and does not introduce the background literature relevant to the questions. Instead, a loose discussion of how pathology may correlate with clinical features in PD and LRRK2-PD is presented. It is advised that the revised introduction be centered on the relevant literature which is largely references 22-24 that deal with the subject at hand. Further, if reference 4 is to be discussed (which it probably should not be), then it would be prudent to also give fair weight to counter-reviews that oppose those interpretations published recently

by Halliday, Surmier, Brundin and others. These could be saved for the discussion or omitted.

Response: The introduction has been revised to contain the relevant background for the work (page 5). Reference 4 has been omitted, instead, a review by Halliday et al. has been included. The text has been revised accordingly (page 4).

Experimental considerations:

1- The LRRK2 Rab35 phosphorylation data (included 'only for reviewers') needs to be presented in this paper. It is unacceptable to be saved for a different publication, for example, and not be included as a supplemental figure, given the focus on Rab35 in this paper.

Response: We understand the reviewer's concern. The finding of Rab35 phosphorylation had made by some of the coauthors of our paper (BD Lee, GR Jeong, JR Bae), and these authors submitted their findings, including the phosphorylation of Rab35 by LRRK2, to another journal as a separate paper. During the course of revision, their paper was published in Molecular Neurodegeneration. The work involves collaboration among many research groups. As much as we would like to have the phosphorylation data in our paper, we are in a situation where we cannot force our wish to have the data included in our paper. Furthermore, Rab35 phosphorylation by LRRK2 has been published in Steger M et al., 2017, eLife, while our paper was under review, which makes this part of our work less critical than before. In the revised manuscript, we added these two references in the text (page 10).

2- In figure 2a, lysates from cells not treated the synuclein-positive media needs to be shown as western blots. These blots may be blank. Also, the antibody should be given in the figure legends (assuming this is the synuclein antibody from the Methods?, but it could also be antibodies to venus..)

Response: Time 0 is the no-treat control sample. This lane has been re-labeled 'no-treat' to make this point clear. The antibody used for these blots has been indicated in the figure legend (page 47).

3- In interpretation of Figure 2, that data could all be explained by enhanced uptake of synuclein mediated by LRRK2. Or, it could be both increased uptake as well as increased release of a-synuclein. The later was not tested but needs to be tested for interpretation. It

could be that LRRK2 upregulates both uptake AND release, this needs to be determined in this model.

Response: As the reviewer pointed out, interpretation of the data in Figure 2 is more complicated than it seems. Figure 2a shows ‘the levels of the internalized a-synuclein’ that remain in the cells after the uptake (representing the steady state, which is the function of uptake and degradation rates), rather than showing the rate of uptake per se. Because the internalized a-synuclein is rapidly delivered to lysosomes and degraded, our simple western analysis cannot be deciphered into the uptake rate. Therefore, we intentionally did not use the word ‘uptake’ in the figure legend. We noticed that we used the word ‘uptake’ in one place in the text (page 8 line 5) in initial submitted manuscript. This has been changed to ‘the steady state levels’ in the revised manuscript.

Secretion is also a rather complicated issue. Because a-synuclein is constantly secreted and internalized AND secreted again, the extracellular a-synuclein in the media is the sum of ‘primary’ secretion and the ‘secondary’ secretion (or internalization-and-resecretion). It is very difficult to measure the ‘primary secretion’. However, we CAN measure the ‘secondary secretion’ by measuring the Venus fluorescence in the media, which represent the co-aggregation between the internalized a-synuclein and the recipient-expressed a-synuclein. This secondary secretion was increased in the cultured cells overexpressing WT LRRK2 and even more increased in the cells expressing G2019S LRRK2. The data is included in Figure 2g and explained in the text (page 9-10) in the revised manuscript.

3- In Figure 3, the co-localization analysis is of very low value. The reason is that the Rab35 expression is mostly diffuse, likely due to over-expressing the protein for too long in the cell lines, many groups express for only a few hours to preserve a proper ratio of membrane-to-cytosolic ratio. Further, the interpretation of a-synuclein 'aggregates' may simply be a-synuclein dimers localized to endosomes. The IPs are much stronger evidence. If the Author's want to keep the analysis, the co-loc should be re-performed under conditions that identify a-synuclein aggregates (e.g., thioflavin), and preserve Rab35 membrane localization, and present convincing photomicrographs (super-resolution may be needed for convincing co-loc).

Response: The purpose of the colocalization experiment is to determine which Rab protein best identifies the endosomal compartment where the transferred a-synuclein is present. We knew from our previous studies as well as others’ that a-synuclein is transferred through endocytosis. Endosomal trafficking follows the endocytic internalization. Since Rab proteins specify the vesicle trafficking pathways and the vesicles involved, our intension was simply to find the Rab proteins that label the vesicles containing the transferred a-synuclein. We reasoned that IP experiment would not work here, because there would be a topological

barrier between Rabs and a-synuclein; Rab proteins bind to the cytosolic side of vesicles, whereas internalized a-synuclein would be present in the lumen of vesicles. For this reason, we decided to use immunofluorescence labeling of a-synuclein and Rabs in the recipient cells.

The transferred a-synuclein are likely monomers and dimer/trimer/oligomers, since these are the species that are found in the media. We failed to detect fibrils in the media, and no published work showed the presence of fibrils in the culture media. Years ago, we published a work showing that fibrillation is tightly linked to the inclusion body-forming process (Lee and Lee, JBC, 2002). Therefore, we believe that the transferred a-synuclein, at pre-inclusion body stage, will not be stained by thioT. We do see thioT-positive inclusion bodies in the recipient cells. However, these inclusion bodies probably are the end-stage product of vesicle trafficking. These inclusion bodies are not of our interest in the experiments described in Figure 3, since we are interested in the trafficking process after the internalization. For these reasons, immunolabeling would be better than thioT staining in detection of the transferred a-synuclein in the recipient cells.

As the reviewer noted, we expressed Rab proteins for over two days; Cells were transduced by lentiviral vectors for 6 h, after which the viruses were washed away, and then, cells were co-cultured with a-synuclein-overexpressing SH-SY5Y cells for 2 days. The reason for the 2-day incubation is that it takes that much time to detect the transferred a-synuclein in the recipient cells. In the revised manuscript, we included the immunofluorescence images of Rabs to show the localization patterns of these proteins (Supplementary Figure 4a and page 11). Rab1 and Rab35 exhibited more diffuse pattern than Rab8. However, vesicular patterns did co-exist for all the Rabs tested. We analyzed these vesicular structures for colocalization measures. And, we replaced the confocal data with much clearer images and added higher magnification images to clarify the colocalization between Rabs and a-synuclein in the recipient cells (Figure 3a and supplementary figure 4b).

Reviewer #2 (Remarks to the Author):

The article by Bae et al "LRRK2 kinase regulates intercellular transmission of α -synuclein 2 aggregates through phosphorylation of Rab35." addresses an issue of great interest to the PD field and more generally to those interested in cell biology. The experiments and the document are generally well composed, and mostly appropriate. I find the use of the array of techniques good, but somewhat disjointed. Although there are many strengths to the work, which I feel should be championed, I feel the writing is massively over-simplified and the conclusions far too broad based on the evidence presented. I am enthusiastic for a lot of the work, and recommend that the editors STRONGLY consider a resubmission, but in its' present state the document is crushed by brevity, over interpretation of the data, and the perceived need to 'tick the boxes' of a novelty-driven, high impact journal. The impact of the work wouldn't be diminished by a more honest interpretation, but might be more appreciated by the audience.

For example (this would be my major concern), throughout the document there is little self-awareness or self-criticism, rather the furthest possible 'important' extrapolation of the data is presented as the simplest interpretation of the results.

An overarching theme for the way the document is written is that Lrk-1 is not just a homologue, but is analogous to LRRK2.

C. elegans (lrk-1) and Drosophila (Lrrk) genes have an ancient origin distinct from that of LRRK2, while perhaps a conserved homolog for mammalian LRRK1 (important during embryogenesis and development in mammals, but absent very soon after birth) there is little evidence of a good relationship between Lrk-1 and LRRK2 which is expressed exclusively (without LRRK1) in adult mammalian tissue. This is not species snobbery from this reviewer, it is more the lack of transparency and overly hyperbolic writing in the document that stimulates such a comment. A more unbiased writing style would improve the reviewers appreciation of the data, rather than the drum-banging.

Response: I, the corresponding author, am deeply embarrassed by not being careful about the point the reviewer has raised and grateful for the opportunity to improve the manuscript by revisiting our interpretation of the data and the writing style. *lrk-1* not being analogous to LRRK2 has been clarified in the revised manuscript (page 6), and thus, we stated the caveat in interpreting the *C. elegans* data in the discussion section (page 17). We also carefully went through the text and changed the writing style/tone to conform to the reviewer's point.

Similarly, much is made of the Lrk-1 hypomorphs/functional knockouts being evidence for LRRK2's role in transmission of synuclein; however, in other reports the worm Lrk-1 mutants have several other alterations (odd axonal arborizations, presumably also synaptic function, axonal proteins localized in dendrites etc. e.g., Sakaguchi-Nakashima 2007) which might impact upon synuclein transmission between cells independently of whatever (the absent) LRRK2 protein might do. Many of these alterations might be due to disturbed neuronal development or general worm health in the absence of a LRRK1/2 homologue, and thus the data should not be used to support an argument based on impairment of 'LRRK2' function that would normally facilitate synuclein transmission between cells (at least without open discussion of other possibilities).

This reviewer doesn't even see these shortfalls as a problem for the current data set, merely the language used in the description of the data.

Response: This is an important point as well. We have discussed the alternative possibilities by which a-synuclein propagation could be altered in *lrk-1* mutant worms (page 17).

Another overall concern, not addressed, is that much of the elegans data is assayed at day 13... when ~20% Lrk-1 and 50% of asyn worms are already dead... are animals dying early because inclusions and synuclein propagation (as inferred) or prior to / in absence of these phenomenon

Response: Inclusion bodies become abundant at day 13, which is why we analyzed the inclusion formation at this time point. Inclusion body itself may not be the cause of the degenerative phenotypes, rather the outcome of aggregate accumulation (here, aggregate being the general term that includes oligomers and other multimers). However, propagation of a-synuclein starts at larvae stage and persists, even increases, during aging. In other words, a-synuclein propagation precedes all the other phenotypes the worm shows in our analyses (including the inclusion formation). This was described in the reference 8 (Kim et al., 2016, Autophagy).

Specific concerns:

1) The introduction makes mention of the fact that most LRRK2 PD cases present with synucleinopathy, yet should also mention that nearly half do not.

Response: As per reviewer's suggestion, introduction was corrected in the revised manuscript (page 5). "Interestingly, a large proportion of PD patients with LRRK2 mutations exhibited α -synuclein positive LBs, even though there are LB-negative LRRK2-PD."

2) It is inappropriate to state that BiFC transgenic C. elegans lines are LRRK2-deficient (lrk-1.) Besides the arguments listed above, at least one report states LRRK2 inhibitors don't work on Lrk-1 (Yao 2013, hum mol gen).

Response: This has been corrected throughout the manuscript.

3) Authors state "These results indicate that LRRK2 plays an important role in α -synuclein transmission in *C. elegans*." asides from LRRK2 not being present in *C. elegans*, *lrk-1* deficient worms not otherwise normal... maturation rates, feeding, movement, among other alterations could account for a reduction in synuclein transmission which is entirely indirect

Response: Extrapolation of the worm *lrk-1* results to the functions of mammalian LRRK2 has been avoided/toned down throughout the manuscript. Alternative, indirect explanation of the results was provided in Discussion (pages 17).

4) Even figure legends and data are labelled LRRK2 inappropriately.

Response: 'LRRK2' was changed to '*lrk-1*' in figure legends as well as in the text.

5) Fig 1 J. Is this drawn from a single example animal / section or is this aggregated imagery composed from several animals? Fig. 1K., number of projections is an odd metric. From the images it isn't as if there is a percentage (others labelled), or knowledge of whether signal is multiple parts of one axon... this might be better quantified as signal area / densitometry analyses?

Response: Figure 1j has been modified in revised manuscript. In revised manuscript, we showed two clearer examples of alpha-synuclein-immunoreactive axons, one in a wild-type rat and the other in a LRRK2-deficient animal. Panels k and l of figure 1 reported new data in a format suggested by this Reviewer. Data represented unbiased measurements of axonal length and density using the Space Balls stereological tool. Measurements were carried out in pontine sections from 9 wild-type and 9 LRRK2-deficient rats. For each animal three separate sections were analyzed.

6) Figure 2. For cell lines infected with LRRK2 it is important to show the levels of overexpression of each construct, also the number of cells and their proximity to each other must be approximately equal to make sense of 'transmission' likelihood?

Response: Western blotting of LRRK2 was included in supplementary figure 3a,b to show the overall expression levels. The overall expression level of each LRRK2 proteins was quantified in supplementary figure 3b. In addition, immunofluorescence images of LRRK2 and the quantification of the expression levels were attached in supplementary figure 3c, d. The image shows that the overall distribution of LRRK2-positive cells is indistinguishable among the experimental groups. These results are described in page 8 of the text.

7) why is there no regular sized synuclein in the blot? Even in insoluble fractions, should be... unless there is none in the cell line? S can the authors also provide soluble? Or are cells internalising aggregated synuclein? It is also unclear from the text what synuclein is "relative" to -

presumably actin control... would be better assessed relative to total synulcein in the cell?

Response: The soluble fraction blot has been attached in the revised manuscript. Soluble fraction blot showed the normal size a-synuclein. After treatment of a-synuclein-containing media, the aggregated forms of a-synuclein accumulated only in the triton-insoluble fractions. In soluble fraction, only LRRK2 G2019S-transduced cells showed the increase in the levels of monomeric a-synuclein. These are described in page 9.

The levels of a-synuclein were normalized with those of actin, thereafter the relative levels of a-synuclein to the maximum levels of the internalized total a-synuclein in mock-transduced SH-SY5Y cells were calculated. The explanation on the quantification was included in the revised manuscript in methods (page 29).

8) *Problems with data presentation: number of cells expressing seems inappropriate... all are 1 yet with variability? Most GS must contain 1.4 with 10% variance with odd cell being way off? Also, n is not 3 surely (as stated)?, n is likely 200 from 3 independent experiments. This needs to be much clearer throughout; change the writing or the presentation (scatter plots with overlaid mean +/- SEM would help)*

Response: Yes, we realize that the data in Figure 2e are very confusing in its current form. The reviewer correctly understood the data that three independent experiment was performed; each experiment with n = 200. In the revised manuscript, graphs in Figure 2d and Figure 2e (in the original manuscript) were combined in Figure 2f with mean/SEM. The Y axis of figure 2f represents the number of BiFC (+) cells out of 200 cells, and the colors represent the number of BiFC fluorescent puncta in BiFC (+) cells. The explanation has been included in the figure legend (page 48).

9) *No explanation of blue signal DAPI? Again to interpret such data we need to know TF efficiency of each, how many cells, how close they are to each other.*

Response: Blue signals indicate the nuclei, which is labeled with Topro3 iodide. The explanation has been included in the figure legend in revised manuscript (page 47, 48, 49). The cells we used in this study stably express a-synuclein-Venus N or C fusion proteins, so a half the cells express a-synuclein-VenusN, the other half a-synuclein-VenusC in co-culture of these cells (Bae et al., 2014, Nature Commun).

10) Need to address why inhibitor only works in OE not mock. Need to check levels of OE, density in the cells, proximity etc? Residual effects not altered by inhibitor... no endogenous LRRK2? ALSO mutant not different from BAC GS - this is not as expected if a kinase-dependent phenomenon, expect expression levels may be different.

Response: The levels of the endogenous LRRK2 is very low in SH-SY5Y cells; we were not able to detect the endogenous LRRK2 by western blotting. This is probably why it is difficult to see the effects of the inhibitor (page 10). The levels of transduced LRRK2 was quantified and attached in the supplementary figure 3. As you will see in this figure, the overall expression levels tend to be variable, however the differences are not statistically significant. The intracellular distribution of LRRK2 proteins were indistinguishable among the experimental groups. Note that we used adenoviral vectors, so most of SV2 cells were infected with the virus, which represent about 50% of the total cells in the co-culture. This is described in page 8. LRRK2 mutation D1994A is an artificial mutation which harbors the catalytically inactive kinase domain. Overexpression of LRRK2 D1994A was used as a negative control for the responses to the kinase inhibitor. Increased propagation of a-synuclein by the overexpression of either LRRK2 WT or LRRK2 G2019S were reversed by the inhibitor treatment, whereas the propagation in the LRRK2 D1994A-transduced cells was not responsive to the inhibitor. These data confirmed that LRRK2 could enhance the propagation a-synuclein in a kinase activity-dependent manner.

11) Fig 3. Colocalisation experiments must be by coefficients (eg pearsons R) puncta analyses and overlap / coloc number are fine, but require extra information of puncta number, size and intensity to make sense of number colocalised. Similarly n is not clerly reported again (see point 8)

Response: As per the reviewer's suggestion, pearson's coefficient was calculated and attached in revised manuscript in Figure 3c (page 11).

12) Does rab35 LRRK2 interaction occurs with non-overexpressed endogenous proteins also? Seems example blots are missing in Fig3 D? Also a casual glance makes it look like MORE

rab35 associated with LRRK2 in WT than GS in c & d (unclear in 3F10 example). This should be quantified as might be driving force for alterations, or at very least addressed in text

Response: We were not able to detect the endogenous LRRK2 in differentiated SH-SY5Y cells, and thus, we could not confirm the interaction between the endogenous LRRK2 and Rab35. Should we try IP in brain extracts??

The Quantified graph of the IP experiment has been attached in the revised manuscript (Figure 3e, g). There is no difference between the WT and GS mutant. However, since IP experiment is only semi-quantitative, we are hesitant to draw any serious conclusion over this result, rather we merely confirm the interaction between these proteins.

13) other than n's being hard to decipher... statistical analyses inappropriate; presented for 2way RM ANOVA yet paired ttest???

Response: As per the reviewer's suggestion, the p values were calculated by one-way ANOVA with tukey's post hoc test, one-way ANOVA with Dunnet's post hoc test, or two-way ANOVA with Dunnet's post hoc test throughout the revised manuscript when appropriate.

14) Figure 4 n again hard to work out. Life span here and other two in WT shows they seem to mostly ALL die on the same day - here is day 16, p day 14, fig5h is 14. Other than Fig 4h is hard to know how much a difference (1day) is due to variability or treatments... scatters and join bars between replicates would help, e.g., in 3 experiments is WT always the same sine different? Is randomness in the 200?

Response: The Mean life span is very reproducible in *C. elegans*. WT worms (N2) have the mean life span of about 16 days. When V1S and SV2 were injected, the mean life span was reduced to about 13-14 days. In Figure 4h and 5h, the control columns represent V1S+SV2 lines in N2 and Rab35 backgrounds, respectively. To clarify this point, we added the V1S+SV2 labels in the figures. Survival curves from all individual experiments are shown in Supplementary Figures 1, 5, and 6 in revised manuscript, so the readers can appreciate the variability of this assay.

15) Figure 6 labelled "transmission". How is this clearly transmission, and not reduced expression in certain cell types? Which cells are expressing, are they assaying in areas which don't express on the transgene?

Response: This experimental model, as explained in Figure 1a, expresses a-synuclein-VenusN and a-synuclein-VenusC in pharyngeal muscle cells and the connected neurons, respectively, using the cell type-specific promoters. We chose the transgenic lines based on the expression levels of a-synuclein, so that all the transgenic lines express similar levels of a-synuclein. Expression levels of a-synuclein in lines are shown in Supplementary Figures 1, 5, and 6 in the revised manuscript. Expression levels of Rab35 proteins are shown in Supplementary Figure 6d in the revised manuscript. The point about the expression levels is described in the text (pages 6, 11, and 12).

16) Figure 6 Puncta analyses... again if percent coloc. need details on number and intensity of both signals. Better with Pearson's. That said rab increases in intensity are interesting so need to show more e.g., puncta size, number etc. rab levels by wb etc.

Response: In Figure 7 in revised manuscript, we have expanded the description of this figure in the Results section and provided additional and more analysis of the levels and size of the Rab35 positive structures (page 13-15). Moreover we have completely re-written this section to provide more clear description of the analysis and comparisons among groups for the various panels.

17) Figure 6. What is the evidence for target engagement in these experiments (not others) e.g., LRRK2 phosphorylation. Also again the lack of detail on statistical analyses are alarming, was a stats form filled in? Where are details of the ANOVA F-values and df, residuals etc throughout?

Response: Indeed we considered the levels of LRRK2 phosphorylation represent an indirect measure of target engagement. We apologize for the lack of details in the statistical analysis. We have completely revise the Methods, Results and Figure legend sections as applicable. We performed one-way ANOVA with post-hoc Dunnet when comparing to non-tg vehicle and with Tukey-Kramer when comparing to the tg treated mice. Moreover we have now provided the detailed F and P values for each of the experiments.

18) Supplemental fig 6. The inhibitor reduces LRRK2 p but also seemingly Rab35 and aSyn levels. While they show the rescue attempt is superseded by Rab constitutively active, these conclusions require evidence for whether LRRK2 protection is mimic by reduction in pRab?

Response: We also find it interesting that the inhibitors cause the reduction in levels of Rab35 and a-synuclein. Reduction in a-synuclein levels is consistent with the IHC data showing the reduction of the total a-synuclein deposition. This seems to be the result of the rescue effect of the kinase inhibitor.

As much as we would like to measure the levels of phospho-Rab35 in the mouse brains, we unfortunately do not have tools to accomplish this at present. Although we tried IP of Rab35 and mass analysis to quantify the phospho-forms, we were not able to detect phospho-Rab35. It is well-known that phosphoproteins are difficult to ionize. It is a technical problem we should overcome. However, it will take us a significant amount of time and effort to solve this problem, which will significantly delay the publication of the work. In the revised manuscript, we stated that measurement of phospho-Rab35 has not been successful in mouse brains and needs to be addressed (page 14).

As for the reduction in Rab35 levels, it could be due to either increased degradation or reduced synthesis of Rab35 protein. The role of LRRK2-mediated Rab35 phosphorylation in the reduction of its levels would be another interesting topic for investigation. As much as it is interesting and important, this issue is better addressed in a separate study in the future.

Reviewer #3 (Remarks to the Author):

The manuscript by Bae and colleagues presents data that “LRRK2 regulates the intercellular transmission of a-synuclein aggregates via phosphorylation of Rab35.” The study explores intercellular transmission of the Parkinson’s disease-associated protein a-synuclein using C.elegans, human SH-SY5Y cells, rats and a-synuclein transgenic mice. The study offers interesting data that LRRK2 plays a role in regulating a-synuclein transmission via the phosphorylation of Rab5. On the other hand, the manuscript is confusing in many places because of the switching between the phrases “a-synuclein transmission” and “a-synuclein aggregate transmission.” The main claim—that a-synuclein aggregates are transmitted from cell-to-cell—is not well-supported by the data. Some specific points to address are as follows.

1) lines 78-104:

“a-synuclein transmission” is used on lines 83-4, 94, 99, and then the paragraph ends by switching to the phrase “a-synuclein aggregate transmission” on line 103.

Why the switch from “a-synuclein transmission” to “a-synuclein aggregate transmission”?

*Between lines 83-102 there is no data that shows that α -synuclein aggregates were transmitted. How was it ruled out that α -synuclein monomers exchanged between the nerve and muscle cells in wild type worms? Perhaps in the *lrk-1* worms the mutation inhibits release of the monomer.*

More on this point, in Figure 1a, the right-hand images of the donor and recipient cells shows that both cells release monomers, which are not aggregates.

Response: Based on the cells studies (e.g., Lee et al., 2005, J. Neurosci; Jang et al., 2010, J Neurochem), both monomers and multimers (dimer/trimer/oligomers) are secreted from cells. Although the templated seeding mechanism postulates that the high-order multimers are the ones to serve as seeds for aggregate transmission, it has not been properly demonstrated. Therefore, as per the reviewer's suggestion, we only used the term 'a-synuclein propagation' in the revised manuscript (see also the comment 1 by reviewer 1).

2) *Figure 2, legend: "(c-e) Alterations in LRRK2 kinase activity regulates cell-to-cell transmission of α -synuclein. VIS cells were co-cultured with either myc conjugated WT, GS, or DA LRRK2 infected SV2 cells. The transmission of α -synuclein aggregates were analyzed by measuring BiFC (+) cell (%)."*

a) *How do you know that α -synuclein aggregates were transferred between donor and recipient cells? Were α -synuclein aggregates detected in the culture media? Were monomers detected in the culture media?*

b) *What does "co-aggregates" mean?*

Response: In the dual cell BiFC co-culture system, the a-synuclein-VenusN and a-synuclein-VenusC proteins are secreted and transferred to the neighboring cells that express either of these proteins. When a-synuclein-VenusN proteins are transferred to the cells expressing a-synuclein-VenusC, and when these proteins 'co-aggregate', the protein complexes emit Venus fluorescence. The 'co-aggregates' referred to as the protein complexes between the transferred a-synuclein-Venus and the recipient-expressed a-synuclein-Venus. The co-aggregates are fluorescent and represent the propagation of a-synuclein between cells. For the details of the system, please refer to Bae et al., 2014, Nature Commun (ref 26).

As explained in the response to the comment 1, cells secrete both monomeric and multimeric a-synuclein. Although the theory assumes that the multimers are the principal species mediating a-synuclein propagation, it has not been thoroughly proved yet. Therefore, we toned down the writing style on 'aggregate transmission' wherever appropriate in the revised manuscript. Nevertheless, we detect Venus fluorescence-positive 'co-aggregates' in the culture media, suggesting that not only the co-aggregation occurs, but also these aggregates

are indeed secreted from cells. The data of secretion of the fluorescence-positive aggregates were included in the page 9-10 in the revised manuscript (Figure 2g).

3) Disconnect between text (lines 55-174) regarding Figure 3 and the legend to Figure 3.

“ α -synuclein transmission” is used four times between lines 55-174. Then a shift occurs in the legend to “ α -synuclein aggregates”: “(e, f) VIS cells were co-cultured with LRRK2 overexpressing SV2 cells which were co-infected with either mock or Rab35 DN. After 3 days of co-culture, transmission and co-aggregation of α -synuclein aggregates were calculated by measuring the BiFC (+) cell (%)”.

This jarring disconnect between the main text and the figure legend. There is no evidence that aggregates were transmitted. Just because BiFC aggregates were detected does not necessarily mean that aggregates were transmitted. How can you rule out that some cells release α -synuclein monomers, which then are endocytosed by neighboring cells, and the increase in the intracellular concentration of monomers causes aggregation inside the recipient cells? Were BiFC aggregates detected in the cell culture medium? Monomers?

Response: Again, the reviewer correctly pointed out that one cannot rule out the possibility of monomer secretion. In revised manuscript, we toned down the previous claim that aggregates are transferred in this system as mentioned in the responses to comments 1 and 2 above. Detection of the BiFC-positive aggregates was presented in Figure 2g in the revised manuscript. This had also been extensively addressed in our previous paper (Ref 26, Bae et al., 2014, Nature Commun).

4) Figure 1 legend, (b) BiFC fluorescence in wild type and *lrk-1(tm1898)* mutant worms at day 13. The red arrows: BiFC signals, the red arrowheads: inclusions in pharynx.

There are no red arrows in the figure; only red arrowheads.

Response: The legend was corrected to ‘The red arrow heads: inclusions in pharynx’ in revised manuscript.

5) Figure 1 legend, (c) *Quantification of BiFC fluorescence at day 13. Twenty worms for each line were used.*

The BiFC fluorescence values in plot c. More information is needed as to how the values (4,000 and 2,000) were obtained. Are you averaging intensities over the entire green portions of the worms?

Response: More information that how the values were obtained was provided in the method section of the revised manuscript (page 24, 25). The images were captured by using standard filters and appropriate lasers, and acquisition parameters were fixed for excitation intensity, gain value, and voltages on each imaging channel. After image acquisition from each transgenic line, the pharynx area was selected by using drawing/selection tools in the FV10-ASW 3.1 viewer (Olympus, Tokyo, Japan). Subsequently, the integrated fluorescence intensity and area of the selected regions of interest (ROIs) within the pharynx were measured using FV10-ASW 3.1 analysis software (Olympus, Tokyo, Japan). The BiFC fluorescence intensity were determined as the relative fluorescence intensity per area (μm^2) in pharynx region. The values represent the average intensities of several worms in each line.

6) *line 155 LRRK2-mediated Rab35 phosphorylation regulates α -synuclein transmission paragraph and figure (Fig. 3)*

You can help the Reader if you state explicitly what cells you are using in this paragraph. For example, in line 167 it says "...in the recipient cells." It is clearer to say "in the SH-SY5Y recipient cells." Same for the legend to Figure 3. Nowhere in the legend does it say what cells are being used.

Response: As per reviewer's suggestion, we corrected 'recipient cells' to 'recipient SH-SY5Y cells' in the revised manuscript (page 11). The cells used in Figure 3 was included in Figure legend in the revised manuscript (page 48). The experimental details about Figure 3 was attached in page 30 as SH-SY5Y co-culture assay in revised manuscript.

7) *Figure 4 a, b. (a, b) BiFC fluorescence in wild-type and rab-35 transgenic models at day 13. White arrows: BiFC signals, red arrowheads: inclusions in pharynx. Twenty worms for each line were used.*

It is not clear how the quantitation was done. How are the average fluorescence values for

“BiFC signals (white arrows)” obtained. In the rab-35 worm (Fig. 4a) there are two white arrows, which point to structures that have very different fluorescence intensities. It looks like there is significant variation in the intensities of these structures. That variation does not seem to be reflected in the relatively small error bar on the rab-35 black bar in Fig. 4b.

Response: The details of BiFC fluorescence quantification is now provided in the method section of the revised manuscript (page 24-25). White arrows actually indicate BiFC signal IN NEURONS. It is our mistake not to mention that the arrows indicate neuronal BiFC. Because we realized that marking the neuronal BiFC only caused confusion, we decided to delete the white arrows from the picture in the revised manuscript. As explained in the methods, we measured the fluorescence in the entire pharynx area. Variations in fluorescence in the pharynx were fairly low as the error bars indicate.

8) Unless better evidence can be obtained that α -synuclein aggregates were transferred intermolecularly, the title should be changed.

Response: The title has now been changed to “LRRK2 kinase regulates propagation of α -synuclein via phosphorylation of Rab35.”

Reviewers' comments:

Reviewer #1 (Remarks to the Author):

All my concerns appear to have been addressed, and I am enthusiastic for the revised version.

Reviewer #2 (Remarks to the Author):

The article by Bae et al "LRRK2 kinase regulates intercellular transmission of α -synuclein 2 aggregates through phosphorylation of Rab35." addresses an issue of great interest to the PD field and more generally to those interested in cell biology. The experiments and the document are generally well composed, and mostly appropriate. I find the use of the array of techniques good, but somewhat disjointed. Although there are many strengths to the work, which I feel should be championed, I feel the writing is massively over-simplified and the conclusions far too broad based on the evidence presented. I am enthusiastic for a lot of the work, and recommend that the editors STRONGLY consider a resubmission, but in its' present state the document is crushed by brevity, over interpretation of the data, and the perceived need to 'tick the boxes' of a novelty-driven, high impact journal. The impact of the work wouldn't be diminished by a more honest interpretation, but might be more appreciated by the audience.

For example (this would be my major concern), throughout the document there is little self-awareness or self-criticism, rather the furthest possible 'important' extrapolation of the data is presented as the simplest interpretation of the results.

An overarching theme for the way the document is written is that Lrk-1 is not just a homologue, but is analogous to LRRK2.

C. elegans (lrk-1) and Drosophila (Lrrk) genes have an ancient origin distinct from that of LRRK2, while perhaps a conserved homolog for mammalian LRRK1 (important during embryogenesis and development in mammals, but absent very soon after birth) there is little evidence of a good relationship between Lrk-1 and LRRK2 which is expressed exclusively (without LRRK1) in adult mammalian tissue. This is not species snobbery from this reviewer, it is more the lack of transparency and overly hyperbolic writing in the document that stimulates such a comment. A more unbiased writing style would improve the reviewers appreciation of the data, rather than the drum-banging.

R2 comments. The authors have clearly gone to some lengths to improve the manuscripts presentation, analysis and interpretation and I will address the responses below after some general comments.

Major Concerns

- 1) The distinction between lrk-1 has been addressed but only partially; there are still several areas of implied overlap, which I assume are accidental at this stage. However, the writing style leads the reader that way, especially as the introduction and beginning of the results lack such explanation. The results suggest some functional overlap, which is important, should be highlighted, AND moreover discussed (although not directly with by LRRK2 overexpression in the worm). Given that lrk-1 may be of an entirely separate lineage to LRRK1 and LRRK2 (LRRK3 has been proposed), and that neither lrk-1 nor LRRK1 contain the N- or C- terminal regions distinct to LRRK2 (which is causal to Parkinson's disease) the differences and similarities are of great interest i.e., are those regions which make LRRK2 distinct crucial for the pathophysiology of PD in humans, do they contribute to age-dependent, regional or cell specific vulnerability that is distinct from the a-syn transmission hypothesis?*
- 2) Other results using a similar approach of bifluorescence in C. elegans have not been cited in the introduction or discussed; this is paramount, as the other results show the polar opposite effect of lrk-1 in neuron to neuron transmission (wherein knock-down*

of lrk-1 enhanced BiFC; Tyson 2017) as opposed to muscle to neuron here.

- 3) *The AAV rat experiment is poorly designed, controlled, presented and analysed. The comparison between LRRK2 KO bought from a supplier and normal rats from a different colony is inappropriate; littermate comparisons should have been used, especially with an assay as nuanced as progressive pathology following a vagal injection. There should be only one variable, genotype. Further, animals from 4wk and 8wk (only 4 each) have been pooled. If this really is an effect of LRRK2 loss upon propagation, then a temporal progression should have been detected... The data adds very little to the manuscript, isn't a technique or approach followed elsewhere, similar results have been published and I feel it distracts the reader from the main message of the other more developed data sets.*
- 4) *Concerns with statistical approaches. All bar charts with two comparisons (genotype x treatment for the mouse figure, or brain region x treatment should be analyzed, as presented, by 2-way ANOVA)*

Response: I, the corresponding author, am deeply embarrassed by not being careful about the point the reviewer has raised and grateful for the opportunity to improve the manuscript by revisiting our interpretation of the data and the writing style. *lrk-1* not being analogous to LRRK2 has been clarified in the revised manuscript (page 6), and thus, we stated the caveat in interpreting the *C. elegans* data in the discussion section (page 17). We also carefully went through the text and changed the writing style/tone to conform to the reviewer's point.

Agreed but not well enough and should be in the introduction, as well as corrected in results e.g., "We validated the role of LRRK2 in the propagation of a synuclein in two independent in vivo models that are deficient in the LRRK2 gene. First we used a C. elegans model in which..."

Similarly, much is made of the Lrk-1 hypomorphs/functional knockouts being evidence for LRRK2's role in transmission of synuclein; however, in other reports the worm Lrk-1 mutants have several other alterations (odd axonal arborizations, presumably also synaptic function, axonal proteins localized in dendrites etc. e.g., Sakaguchi-Nakashima 2007) which might impact upon synuclein transmission between cells independently of whatever (the absent) LRRK2 protein might do. Many of these alterations might be due to disturbed neuronal development or general worm health in the absence of a LRRK1/2 homologue, and thus the data should not be used to support an argument based on impairment of 'LRRK2' function that would normally facilitate synuclein transmission between cells (at least without open discussion of other possibilities).

This reviewer doesn't even see these shortfalls as a problem for the current data set, merely the language used in the description of the data.

Response: This is an important point as well. We have discussed the alternative possibilities by which a-synuclein propagation could be altered in *lrk-1* mutant worms (page 17).

*This is to be commended, but is insufficient. As an aside here, the English in many of the revised sections is poor, and I recommend that the corresponding author have the revisions checked for grammatical errors prior to any future submission e.g., “*lrk-1* gene have” “Relationship between is not clear”. Scientifically, the data show that the *Rab35* mutant is similarly protective to the *lrk-1* mutant, in terms of the synuclein effects. Does this ‘phenocopy’ also extend to the other alterations previously described in *lrk-1* mutants (odd axonal arborizations, presumably also synaptic function, axonal proteins localized in dendrites etc. e.g., Sakaguchi-Nakashima 2007). If so this suggests almost complete replication and functional convergence, and if not suggests the phenocopy is distinct to synuclein propagation. Very important findings?*

*Another overall concern, not addressed, is that much of the *elegans* data is assayed at day 13... when ~20% *Lrk-1* and 50% of *asyn* worms are already dead... are animals dying early because inclusions and synuclein propagation (as inferred) or prior to / in absence of these phenomenon*

Response: Inclusion bodies become abundant at day 13, which is why we analyzed the inclusion formation at this time point. Inclusion body itself may not be the cause of the degenerative phenotypes, rather the outcome of aggregate accumulation (here, aggregate being the general term that includes oligomers and other multimers). However, propagation of a-synuclein starts at larvae stage and persists, even increases, during aging. In other words, a-synuclein propagation precedes all the other phenotypes the worm shows in our analyses (including the inclusion formation). This was described in the reference 8 (Kim et al., 2016, Autophagy).

*Obviously propagation precedes inclusion formation & blebbing. The question is the relationship between increased mortality and these measures. As many worms have died prior to the day at which the aggregation (inclusion) was assessed, the figure of ~20% of worms with inclusions on day13 may be an underestimate, in that the 20% of dead worms by that stage may have had inclusions before death. There is also the question of the relationship between inclusions, blebs and axonal fragmentation... ~30% of worms show a bleb, 12 % fragmented, and ~20% inclusion... so are all affected worms showing a bleb with 2/3 of them also having an inclusion and 1/3 fragmented or all these separate? The suggestion that inclusion percentage is reduced in the *lrk-1* more than bleb or fragmentation suggests these are distinct cellular phenotypes; it would be good to know.*

Specific concerns:

1) *The introduction makes mention of the fact that most LRRK2 PD cases present with synucleinopathy, yet should also mention that nearly half do not.*

Response: As per reviewer's suggestion, introduction was corrected in the revised manuscript (page 5). "Interestingly, a large proportion of PD patients with LRRK2 mutations exhibited α -synuclein positive LBs, even though there are LB-negative LRRK2-PD."

Poor choice of wording; it isn't a large portion vs a bit; most estimates put synuclein aggregates in LRRK2 (G2019s) at ~50:50.

2) *It is inappropriate to state that BiFC transgenic C. elegans lines are LRRK2-deficient (lrk-1.) Besides the arguments listed above, at least one report states LRRK2 inhibitors don't work on Lrk-1 (Yao 2013, hum mol gen).*

Response: This has been corrected throughout the manuscript.

Not well enough and the issue of LRRK2 inhibitor being ineffective against lrk-1 was not.

3) *Authors state "These results indicate that LRRK2 plays an important role in α -synuclein transmission in C. elegans." besides from LRRK2 not being present in C. elegans, lrk-1 deficient worms not otherwise normal... maturation rates, feeding, movement, among other alterations could account for a reduction in synuclein transmission which is entirely indirect*

Response: Extrapolation of the worm lrk-1 results to the functions of mammalian LRRK2 has been avoided/toned down throughout the manuscript. Alternative, indirect explanation of the results was provided in Discussion (pages 17).

Not well enough

4) *Even figure legends and data are labelled LRRK2 inappropriately.*

Response: 'LRRK2' was changed to 'lrk-1' in figure legends as well as in the text.

Not well enough

5) Fig 1 J. Is this drawn from a single example animal / section or is this aggregated imagery composed from several animals? Fig. 1K., number of projections is an odd metric. From the images it isn't as if there is a percentage (others labelled), or knowledge of whether signal is multiple parts of one axon... this might be better quantified as signal area / densitometry analyses?

Response: Figure 1j has been modified in revised manuscript. In revised manuscript, we showed two clearer examples of alpha-synuclein-immunoreactive axons, one in a wild-type rat and the other in a LRRK2-deficient animal. Panels k and l of figure 1 reported new data in a format suggested by this Reviewer. Data represented unbiased measurements of axonal length and density using the Space Balls stereological tool. Measurements were carried out in pontine sections from 9 wild-type and 9 LRRK2-deficient rats. For each animal three separate sections were analyzed.

Comments on this data set are included above

6) Figure 2. For cell lines infected with LRRK2 it is important to show the levels of overexpression of each construct, also the number of cells and their proximity to each other must be approximately equal to make sense of 'transmission' likelihood?

Response: Western blotting of LRRK2 was included in supplementary figure 3a,b to show the overall expression levels. The overall expression level of each LRRK2 proteins was quantified in supplementary figure 3b. In addition, immunofluorescence images of LRRK2 and the quantification of the expression levels were attached in supplementary figure 3c, d. The image shows that the overall distribution of LRRK2-positive cells is indistinguishable among the experimental groups. These results are described in page 8 of the text.

OK

7) why is there no regular sized synuclein in the blot? Even in insoluble fractions, should be... unless there is none in the cell line? S can the authors also provide soluble? Or are cells internalising aggregated synuclein? It is also unclear from the text what synuclein is "relative" to -

presumably actin control... would be better assessed relative to total synulcein in the cell?

Response: The soluble fraction blot has been attached in the revised manuscript. Soluble fraction blot showed the normal size a-synuclein. After treatment of a-synuclein-containing media, the aggregated forms of a-synuclein accumulated only in the triton-insoluble fractions.

In soluble fraction, only LRRK2 G2019S-transduced cells showed the increase in the levels of monomeric α -synuclein. These are described in page 9.

The levels of α -synuclein were normalized with those of actin, thereafter the relative levels of α -synuclein to the maximum levels of the internalized total α -synuclein in mock-transduced SH-SY5Y cells were calculated. The explanation on the quantification was included in the revised manuscript in methods (page 29).

OK, but there is still the need to explain why there is no monomeric synuclein in the insoluble fraction? See Volpicelli-Daley 2011.

Could it be that the triton insoluble fraction here is detecting only very high molecular weight aggregates (>100kd) that were only in the media, not the cells? A western blot of the secreted α -syn media alone may prove useful; I'd be very surprised and concerned if there were no soluble synuclein asides very high molecular weight aggregates?

8) Problems with data presentation: number of cells expressing seems inappropriate... all are 1 yet with variability? Most GS must contain 1.4 with 10% variance with odd cell being way off? Also, n is not 3 surely (as stated)?, n is likely 200 from 3 independent experiments. This needs to be much clearer throughout; change the writing or the presentation (scatter plots with overlaid mean +/- SEM would help)

Response: Yes, we realize that the data in Figure 2e are very confusing in its current form. The reviewer correctly understood the data that three independent experiment was performed; each experiment with $n = 200$. In the revised manuscript, graphs in Figure 2d and Figure 2e (in the original manuscript) were combined in Figure 2f with mean/SEM. The Y axis of figure 2f represents the number of BiFC (+) cells out of 200 cells, and the colors represent the number of BiFC fluorescent puncta in BiFC (+) cells. The explanation has been included in the figure legend (page 48).

Throughout this seems unclear, if 200 cells or worms (technical replicates) were compressed to a single datum for each independent replicate (x3) on would expect greater variation? Scatter plots instead of bar charts would help understand variance or not in the data and how statistics were performed (likewise F values reported with degrees of freedom and residuals)

9) No explanation of blue signal DAPI? Again to interpret such data we need to know TF efficiency of each, how many cells, how close they are to each other.

Response: Blue signals indicate the nuclei, which is labeled with Topro3 iodide. The explanation has been included in the figure legend in revised manuscript (page 47, 48, 49).

The cells we used in this study stably express a-synuclein-Venus N or C fusion proteins, so a half the cells express a-synuclein-VenusN, the other half a-synuclein-VenusC in co-culture of these cells (Bae et al., 2014, Nature Commun).

OK

10) Need to address why inhibitor only works in OE not mock. Need to check levels of OE, density in the cells, proximity etc? Residual effects not altered by inhibitor... no endogenous LRRK2? ALSO mutant not different from BAC GS - this is not as expected if a kinase-dependent phenomenon, expect expression levels may be different.

Response: The levels of the endogenous LRRK2 is very low in SH-SY5Y cells; we were not able to detect the endogenous LRRK2 by western blotting. This is probably why it is difficult to see the effects of the inhibitor (page 10). The levels of transduced LRRK2 was quantified and attached in the supplementary figure 3. As you will see in this figure, the overall expression levels tend to be variable, however the differences are not statistically significant. The intracellular distribution of LRRK2 proteins were indistinguishable among the experimental groups. Note that we used adenoviral vectors, so most of SV2 cells were infected with the virus, which represent about 50% of the total cells in the co-culture. This is described in page 8. LRRK2 mutation D1994A is an artificial mutation which harbors the catalytically inactive kinase domain. Overexpression of LRRK2 D1994A was used as a negative control for the responses to the kinase inhibitor. Increased propagation of a-synuclein by the overexpression of either LRRK2 WT or LRRK2 G2019S were reversed by the inhibitor treatment, whereas the propagation in the LRRK2 D1994A-transduced cells was not responsive to the inhibitor. These data confirmed that LRRK2 could enhance the propagation a-synuclein in a kinase activity-dependent manner.

OK

11) Fig 3. Colocalisation experiments must be by coefficients (eg pearsons R) puncta analyses and overlap / coloc number are fine, but require extra information of puncta number, size and intensity to make sense of number colocalised. Similarly n is not clerly reported again (see point 8)

Response: As per the reviewer's suggestion, pearson's coefficient was calculated and attached in revised manuscript in Figure 3c (page 11).

OK

12) Does rab35 LRRK2 interaction occurs with non-overexpressed endogenous proteins also? Seems example blots are missing in Fig3 D? Also a casual glance makes it look like MORE rab35 associated with LRRK2 in WT than GS in c & d (unclear in 3F10 example). This should be quantified as might be driving force for alterations, or at very least addressed in text

Response: We were not able to detect the endogenous LRRK2 in differentiated SH-SY5Y cells, and thus, we could not confirm the interaction between the endogenous LRRK2 and Rab35. Should we try IP in brain extracts??

The Quantified graph of the IP experiment has been attached in the revised manuscript (Figure 3e, g). There is no difference between the WT and GS mutant. However, since IP experiment is only semi-quantitative, we are hesitant to draw any serious conclusion over this result, rather we merely confirm the interaction between these proteins.

OK

13) other than n's being hard to decipher... statistical analyses inappropriate; presented for 2way RM ANOVA yet paired ttest???

Response: As per the reviewer's suggestion, the p values were calculated by one-way ANOVA with tukey's post hoc test, one-way ANOVA with Dunnet's post hoc test, or two-way ANOVA with Dunnet's post hoc test throughout the revised manuscript when appropriate.

Incorrect. As stated above, graphs presented as 2way analysed by 1 way. Also one should not report two types of post test. Also may not be necessary if interactions are proven significant by 2way ANOVA e.g., genotype x treatment in fig 7.

14) Figure 4 n again hard to work out. Life span here and other two in WT shows they seem to mostly ALL die on the same day - here is day 16, p day 14, fig5h is 14. Other than Fig 4h is hard to know how much a difference (1day) is due to variability or treatments... scatters and join bars between replicates would help, e.g., in 3 experiments is WT always the same sine different? Is randomness in the 200?

Response: The Mean life span is very reproducible in *C. elegans*. WT worms (N2) have the mean life span of about 16 days. When V1S and SV2 were injected, the mean life span was reduced to about 13-14 days. In Figure 4h and 5h, the control columns represent V1S+SV2 lines in N2 and Rab35 backgrounds, respectively. To clarify this point, we added the V1S+SV2 labels in the figures. Survival curves from all individual experiments are shown in Supplementary Figures 1, 5, and 6 in revised manuscript, so the readers can appreciate the variability of this assay.

Scatter plots with replicate joins would be much more clear

15) Figure 6 labelled "transmission". How is this clearly transmission, and not reduced expression in certain cell types? Which cells are expressing, are they assaying in areas which don't express on the transgene?

Response: This experimental model, as explained in Figure 1a, expresses a-synuclein-VenusN and a-synuclein-VenusC in pharyngeal muscle cells and the connected neurons, respectively, using the cell type-specific promoters. We chose the transgenic lines based on the expression levels of a-synuclein, so that all the transgenic lines express similar levels of a-synuclein. Expression levels of a-synuclein in lines are shown in Supplementary Figures 1, 5, and 6 in the revised manuscript. Expression levels of Rab35 proteins are shown in Supplementary Figure 6d in the revised manuscript. The point about the expression levels is described in the text (pages 6, 11, and 12).

OK

16) Figure 6 Puncta analyses... again if percent coloc. need details on number and intensity of both signals. Better with Pearson's. That said rab increases in intensity are interesting so need to show more e.g., puncta size, number etc. rab levels by wb etc.

Response: In Figure 7 in revised manuscript, we have expanded the description of this figure in the Results section and provided additional and more analysis of the levels and size of the Rab35 positive structures (page 13-15). Moreover we have completely re-written this section to provide more clear description of the analysis and comparisons among groups for the various panels.

OK

17) Figure 6. What is the evidence for target engagement in these experiments (not others) e.g., LRRK2 phosphorylation. Also again the lack of detail on statistical analyses are alarming, was a stats form filled in? Where are details of the ANOVA F-values and df, residuals etc throughout?

Response: Indeed we considered the levels of LRRK2 phosphorylation represent an indirect measure of target engagement. We apologize for the lack of details in the statistical analysis. We have completely revise the Methods, Results and Figure legend sections as applicable. We performed one-way ANOVA with post-hoc Dunnet when comparing to non-tg vehicle and with Tukey-Kramer when comparing to the tg treated mice. Moreover we have now provided the detailed F and P values for each of the experiments.

OK but not fully. I'd also request scatter plots as they are much more informative of technical v independent replication. This is the strongest part of the manuscript, but again a 2way ANOVA should have been employed

18) Supplemental fig 6. The inhibitor reduces LRRK2 p but also seemingly Rab35 and aSyn levels. While they show the rescue attempt is superseded by Rab constitutively active, these conclusions require evidence for whether LRRK2 protection is mimic by reduction in pRab?

Response: We also find it interesting that the inhibitors cause the reduction in levels of Rab35 and a-synuclein. Reduction in a-synuclein levels is consistent with the IHC data showing the reduction of the total a-synuclein deposition. This seems to be the result of the rescue effect of the kinase inhibitor.

As much as we would like to measure the levels of phospho-Rab35 in the mouse brains, we unfortunately do not have tools to accomplish this at present. Although we tried IP of Rab35 and mass analysis to quantify the phospho-forms, we were not able to detect phospho-Rab35. It is well-known that phosphoproteins are difficult to ionize. It is a technical problem we should overcome. However, it will take us a significant amount of time and effort to solve this problem, which will significantly delay the publication of the work. In the revised manuscript, we stated that measurement of phospho-Rab35 has not been successful in mouse brains and needs to be addressed (page 14).

As for the reduction in Rab35 levels, it could be due to either increased degradation or reduced synthesis of Rab35 protein. The role of LRRK2-mediated Rab35 phosphorylation in the reduction of its levels would be another interesting topic for investigation. As much as it is interesting and important, this issue is better addressed in a separate study in the future.

OK – but elaborated discussion of these facts would help.

Reviewer #3 (Remarks to the Author):

Below are comments from Reviewer #1 and Reviewer #3 pertaining to the original manuscript.

Reviewer #1

The term 'intercellular transmission', for example in the title and in the text, is verbose compared to the term commonly used in the field- 'propagation'. Further, the word 'transmission' may not be very accurate, because the aggregates themselves may not 'transmit' as it may be assumed in this work (yet not demonstrated). The authors and many others can clearly show a-synuclein inclusion propagation through the brain in rats and mice and in vitro. It is advised to revise the relevant portions of text.

Au Response: As per reviewer's suggestion, the term 'intercellular transmission' has been changed to 'propagation' throughout the text. The title is now 'LRRK2 kinase regulates propagation of a-synuclein via phosphorylation of rab35.'

Reviewer #3

Between lines 83-102 there is no data that shows that a-synuclein aggregates were transmitted. How was it ruled out that a-synuclein monomers exchanged between the nerve and muscle cells in wild type worms? Perhaps in the Irk-1 worms the mutation inhibits release of the monomer. More on this point, in Figure 1a, the right-hand images of the donor and recipient cells shows that both cells release monomers, which are not aggregates.

Response: Based on the cells studies (e.g., Lee et al., 2005, J. Neurosci; Jang et al., 2010, JNeurochem), both monomers and multimers (dimer/trimer/oligomers) are secreted from cells. Although the templated seeding mechanism postulates that the high-order multimers are the ones to serve as seeds for aggregate transmission, it has not been properly demonstrated.

Revised manuscript:

Reviewers 1 and 3 brought up in essence the same issue, i.e., "the aggregates themselves may not transmit" and "how was it ruled out that a-syn monomers [and not aggregates] exchange between nerve and muscle cells?" But the authors did not adequately address this issue.

The worm and SY-SH5Y data can be explained by V1S-a-syn monomers being secreted by one group of cells in the co-culture and then being taken up by cells expressing a-syn-SV2 monomers, or vice versa. Additionally, one cell type could release V1S-a-syn monomers and the other cells in the co-culture could release a-syn-SV2 monomers, resulting in the generation of Venus fluorescence in the media from the V1S-a-syn/a-syn-SV2 dimer. The data in Figure 2 especially, and the related text, would greatly benefit from the authors explicitly stating that their data can also be explained by monomer propagation between cells. The authors have not proven in their revised manuscript that a-syn aggregates are the main species that transmit/propagate from one cell to another.

Reviewer #2 (Remarks to the Author):

The article by Bae et al "LRRK2 kinase regulates intercellular transmission of α -synuclein aggregates through phosphorylation of Rab35." addresses an issue of great interest to the PD field and more generally to those interested in cell biology. The experiments and the document are generally well composed, and mostly appropriate. I find the use of the array of techniques good, but somewhat disjointed. Although there are many strengths to the work, which I feel should be championed, I feel the writing is massively over-simplified and the conclusions far too broad based on the evidence presented. I am enthusiastic for a lot of the work, and recommend that the editors STRONGLY consider a resubmission, but in its' present state the document is crushed by brevity, over interpretation of the data, and the perceived need to 'tick the boxes' of a novelty-driven, high impact journal. The impact of the work wouldn't be diminished by a more honest interpretation, but might be more appreciated by the audience. For example (this would be my major concern), throughout the document there is little selfawareness or self-criticism, rather the furthest possible 'important' extrapolation of the data is presented as the simplest interpretation of the results. An overarching theme for the way the document is written is that Lrk-1 is not just a homologue, but is analogous to LRRK2. *C. elegans* (*lrk-1*) and *Drosophila* (*Lrrk*) genes have an ancient origin distinct from that of LRRK2, while perhaps a conserved homolog for mammalian LRRK1 (important during embryogenesis and development in mammals, but absent very soon after birth) there is little evidence of a good relationship between Lrk-1 and LRRK2 which is expressed exclusively (without LRRK1) in adult mammalian tissue. This is not species snobbery from this reviewer, it is more the lack of transparency and overly hyperbolic writing in the document that stimulates such a comment. A more unbiased writing style would improve the reviewers appreciation of the data, rather than the drum-banging.

R2 comments. The authors have clearly gone to some lengths to improve the manuscripts presentation, analysis and interpretation and I will address the responses below after some general comments.

Major Concerns

*1) The distinction between *lrk-1* has been addressed but only partially; there are still several areas of implied overlap, which I assume are accidental at this stage. However, the writing style leads the reader that way, especially as the introduction and beginning of the results lack such explanation. The results suggest some functional overlap, which is important, should be highlighted, AND moreover discussed (although not directly with by LRRK2 overexpression in the worm). Given that *lrk-1* may be of an entirely separate lineage to LRRK1 and LRRK2 (LRRK3 has been proposed), and that neither *lrk-1* nor LRRK1 contain the N- or C- terminal regions distinct to LRRK2 (which is causal to Parkinson's disease) the differences and similarities are of great interest i.e., are those regions which make LRRK2 distinct crucial for the pathophysiology of PD in humans, do they contribute to age-dependent, regional or cell specific vulnerability that is distinct from the *a-syn* transmission hypothesis?*

Response: We have addressed the reviewer's concerns on the terminology use and interpretation of data in Introduction, Results, and Discussion. We re-wrote the text with great deal of care not to over-interpret the worm data and try as much as we can to separately describe the worm and mammal data. We included the explanation on isotypes and ontology of LRRK genes in the introduction (page 5 and supplementary figure 1) and discussed the

potential role of the N-terminal region in the pathogenesis of PD (page 19).

2) Other results using a similar approach of bifluorescence in C. elegans have not been cited in the introduction or discussed; this is paramount, as the other results show the polar opposite effect of lrk-1 in neuron to neuron transmission (wherein knock-down of lrk-1 enhanced BiFC; Tyson 2017) as opposed to muscle to neuron here.

Response: We cited the Tyson 2017 paper (reference #23) in the introduction (page 5) and extensively discussed in the discussion (pages 19-20).

3) The AAV rat experiment is poorly designed, controlled, presented and analysed. The comparison between LRRK2 KO bought from a supplier and normal rats from a different colony is inappropriate; littermate comparisons should have been used, especially with an assay as nuanced as progressive pathology following a vagal injection. There should be only one variable, genotype. Further, animals from 4wk and 8wk (only 4 each) have been pooled. If this really is an effect of LRRK2 loss upon propagation, then a temporal progression should have been detected... The data adds very little to the manuscript, isn't a technique or approach followed elsewhere, similar results have been published and I feel it distracts the reader from the main message of the other more developed data sets.

Response: (a) We apologize for the lack of clarity concerning the use/source of KO and WT rats. We used homozygous LRRK2 KO rats generated on the Charles River Long-Evans background strain and, as controls, age- and gender-matched wild-types from the same (i.e. Charles River Long-Evans) original colony and vendor. Changes in the new version of the manuscript (Results section, page 8, and Methods section, page 30) emphasize that WT and KO animals, albeit not littermates, shared the same background strain. This issue is also discussed on page 18 as a potential caveat for data interpretation. Finally, it is worth mentioning that earlier papers in which this LRRK2 KO model was characterized have also utilized the same WT controls as in our current investigation (see, for example: Baptista et al., PLoS One, 2013; Davies et al., Biochem J, 2013; Daher et al., PNAS, 2014; Boddu et al., Hum Mol Genet, 2015; West et al., J Comp Neurol, 2014).

(b) In the model of alpha-synuclein propagation triggered by protein overexpression, spreading is affected by neuronal viability in the dorsal motor nucleus of the vagus nerve (DMnX) as well as the efficiency of AAV-induced transduction. When designing this study, we took this in consideration and elected to compare WT and KO rats that had no neuronal loss in the DMnX and a similar percentage of DMnX neurons overexpressing human alpha-synuclein. Based on these strict requirements, comparisons were made between a total of 8 WT and 8 KO rats. Animals were sacrificed at two different time points, i.e. 8 (4 rats/group) and 12 (4 rats/group) weeks. However, in order to enhance statistical power, for some measurements, values at 8 and 12 weeks were pooled and analyzed together. We have now clarified this experimental plan and analytical approach in the Results section (page 8-9) and in the legends for figure 1 (page 50-51) and supplementary figure 3 (page 60-61).

(c) When spreading of alpha-synuclein was compared in pontine sections at 8 vs. 12 weeks post AAV treatment, a time-dependent progression was seen. This observation, which is now reported in the new panel j of figure 1, is in agreement with this Reviewer's prediction and

with findings of earlier studies using this model (see Results section, page 9).

(d) The model of overexpression-induced spreading of alpha-synuclein is particularly suitable for *in vivo* studies looking at neuron-to-neuron propagation of alpha-synuclein. In this model, the spreading protein is generated within donor neurons and, based on anatomical considerations, detection of human alpha-synuclein in regions rostral to the medulla oblongata can only result from at least one trans-synaptic passage of the protein. Overexpression-induced spreading has been extensively characterized and recently used to test therapeutic approaches against alpha-synuclein propagation (Spencer et al., *Acta Neuropathol Commun*, 2017, doi: 10.1186/s40478-016-0410-8). Data generated using this model as part of the present study are original. We also believe they are important as they complement findings in *C elegans* and strengthen their relevance for pathophysiological processes in adult mammalian brain tissue. Complementarity and relevance of the rodent work has been clarified and emphasized in the Discussion section of the revised manuscript (see, for example, page 20).

4) Concerns with statistical approaches. All bar charts with two comparisons (genotype x treatment for the mouse figure, or brain region x treatment should be analyzed, as presented, by 2-way ANOVA)

Response: As per reviewer's suggestion, graphs in figure 7 and supplementary figure 10 were analyzed by using two way ANOVA with Tukey's post hoc test. P values and F values were included in figure legends (pages 57-58, 64-65).

Response: I, the corresponding author, am deeply embarrassed by not being careful about the point the reviewer has raised and grateful for the opportunity to improve the manuscript by revisiting our interpretation of the data and the writing style. *lrk-1* not being analogous to LRRK2 has been clarified in the revised manuscript (page 6), and thus, we stated the caveat in interpreting the *C. elegans* data in the discussion section (page 17). We also carefully went through the text and changed the writing style/tone to conform to the reviewer's point.

Agreed but not well enough and should be in the introduction, as well as corrected in results e.g., "We validated the role of LRRK2 in the propagation of a synuclein in two independent in vivo models that are deficient in the LRRK2 gene. First we used a C. elegans model in which..."

Response: We explained this issue in the introduction (page 5) and carefully went through the text again and make sure that there is no mistake left.

Similarly, much is made of the Lrk-1 hypomorphs/functional knockouts being evidence for LRRK2's role in transmission of synuclein; however, in other reports the worm Lrk-1 mutants have several other alterations (odd axonal arborizations, presumably also synaptic function, axonal proteins localized in dendrites etc. e.g., Sakaguchi-Nakashima 2007) which might impact upon synuclein transmission between cells independently of whatever (the absent) LRRK2 protein might do. Many of these alterations might be due to disturbed neuronal development or general worm health in the absence of a LRRK1/2 homologue, and thus the

data should not be used to support an argument based on impairment of 'LRRK2' function that would normally facilitate synuclein transmission between cells (at least without open discussion of other possibilities). This reviewer doesn't even see these shortfalls as a problem for the current data set, merely the language used in the description of the data.

Response: This is an important point as well. We have discussed the alternative possibilities by which a-synuclein propagation could be altered in *lrk-1* mutant worms (page 17).

This is to be commended, but is insufficient. As an aside here, the English in many of the revised sections is poor, and I recommend that the corresponding author have the revisions checked for grammatical errors prior to any future submission e.g., “lrk-1 gene have” “Relationship between is not clear”. Scientifically, the data show that the Rab35 mutant is similarly protective to the lrk-1 mutant, in terms of the synuclein effects. Does this ‘phenocopy’ also extend to the other alterations previously described in lrk-1 mutants (odd axonal arborizations, presumably also synaptic function, axonal proteins localized in dendrites etc. e.g., Sakaguchi-Nakashima 2007). If so this suggests almost complete replication and functional convergence, and if not suggests the phenocopy is distinct to synuclein propagation. Very important findings?

Response: We realized that the term ‘phenocopy’ was too broad a word given the scope of our analysis. We have changed it to ‘similar effects’ in page 13. We also clarified the fact that the other phenotypes of *lrk-1* mutation had not been analyzed in *rab-35* mutants in page 19. The entire manuscript has very carefully gone through for English editing.

Another overall concern, not addressed, is that much of the elegans data is assayed at day 13... when ~20% Lrk-1 and 50% of asyn worms are already dead... are animals dying early because inclusions and synuclein propagation (as inferred) or prior to / in absence of these phenomenon

Response: Inclusion bodies become abundant at day 13, which is why we analyzed the inclusion formation at this time point. Inclusion body itself may not be the cause of the degenerative phenotypes, rather the outcome of aggregate accumulation (here, aggregate being the general term that includes oligomers and other multimers). However, propagation of a-synuclein starts at larvae stage and persists, even increases, during aging. In other words, a-synuclein propagation precedes all the other phenotypes the worm shows in our analyses (including the inclusion formation). This was described in the reference 8 (Kim et al., 2016, Autophagy).

Obviously propagation precedes inclusion formation & blebbing. The question is the relationship between increased mortality and these measures. As many worms have died prior to the day at which the aggregation (inclusion) was assessed, the figure of ~20% of worms with inclusions on day13 may be an underestimate, in that the 20% of dead worms by that stage may have had inclusions before death. There is also the question of the relationship between inclusions, blebs and axonal fragmentation... ~30% of worms show a bleb, 12 % fragmented, and ~20% inclusion... so are all affected worms showing a bleb with 2/3 of them also having an inclusion and 1/3 fragmented or all these separate? The

*suggestion that inclusion percentage is reduced in the *lrk-1* more than bleb or fragmentation suggests these are distinct cellular phenotypes; it would be good to know.*

Response: We agree with the reviewer's suggestion that the figure of 20% of worms with inclusions on day 13 may be an underestimate for the reason that the reviewer pointed out. In the revised manuscript, this has been clarified in page 20. As for the relationship between inclusion formation, nerve blebs and fragmentation, and worm's mortality, the reviewer raised an important issue. Unfortunately, in the present study, we did not analyze the co-occurrence of these phenotypes. We suspect that the inclusion bodies may be an independent phenotype from axonal pathology and increased mortality, as the worms with inclusions were less frequent than those with the axonal phenotype. This has been discussed in page 20.

Specific concerns:

1) The introduction makes mention of the fact that most LRRK2 PD cases present with synucleinopathy, yet should also mention that nearly half do not.

Response: As per reviewer's suggestion, introduction was corrected in the revised manuscript (page 5). "Interestingly, a large proportion of PD patients with LRRK2 mutations exhibited α -synuclein positive LBs, even though there are LB-negative LRRK2-PD."

Poor choice of wording; it isn't a large portion vs a bit; most estimates put synuclein aggregates in LRRK2 (G2019s) at ~50:50.

Response: The sentence has been changed to the following (page 5). "many PD patients with LRRK2 (G2019S) mutations exhibited α -synuclein-positive LBs^{9,16,17}, even though nearly half the LRRK2(G2019S)-PD cases are LB-negative^{18,19}."

*2) It is inappropriate to state that BiFC transgenic *C. elegans* lines are LRRK2-deficient (*lrk-1*.) Besides the arguments listed above, at least one report states LRRK2 inhibitors don't work on *Lrk-1* (Yao 2013, hum mol gen).*

Response: This has been corrected throughout the manuscript.

*Not well enough and the issue of LRRK2 inhibitor being ineffective against *lrk-1* was not.*

Response: We have gone through the manuscript very carefully to avoid indicating *lrk-1* as worm LRRK2. The results of Yao et al. describing some of the LRRK2 inhibitors did not work for *lrk-1* have been cited in page 5 (reference #14).

*3) Authors state "These results indicate that LRRK2 plays an important role in α -synuclein transmission in *C. elegans*." besides from LRRK2 not being present in *C. elegans*, *lrk-1* deficient worms not otherwise normal... maturation rates, feeding, movement, among other alterations could account for a reduction in synuclein transmission which is entirely indirect*

Response: Extrapolation of the worm *lrk-1* results to the functions of mammalian LRRK2 has been avoided/toned down throughout the manuscript. Alternative, indirect explanation of the results was provided in Discussion (pages 17).

Not well enough

Response: Interpretation of worm data has been revised to make sure that the manuscript does not give the wrong impression of *lrk-1* being analogous to LRRK2.

4) Even figure legends and data are labelled LRRK2 inappropriately.

Response: 'LRRK2' was changed to '*lrk-1*' in figure legends as well as in the text.

Not well enough

Response: We have gone through the manuscript again and corrected all the mistakes.

5) Fig 1 J. Is this drawn from a single example animal / section or is this aggregated imagery composed from several animals? Fig. 1K., number of projections is an odd metric. From the images it isn't as if there is a percentage (others labelled), or knowledge of whether signal is multiple parts of one axon... this might be better quantified as signal area / densitometry analyses?

Response: Figure 1j has been modified in revised manuscript. In revised manuscript, we showed two clearer examples of alpha-synuclein-immunoreactive axons, one in a wild-type rat and the other in a LRRK2-deficient animal. Panels k and l of figure 1 reported new data in a format suggested by this Reviewer. Data represented unbiased measurements of axonal length and density using the Space Balls stereological tool. Measurements were carried out in pontine sections from 9 wild-type and 9 LRRK2-deficient rats. For each animal three separate sections were analyzed.

Comments on this data set are included above

Response: Please see pages 2-3 in this response letter.

6) Figure 2. For cell lines infected with LRRK2 it is important to show the levels of overexpression of each construct, also the number of cells and their proximity to each other must be approximately equal to make sense of 'transmission' likelihood?

Response: Western blotting of LRRK2 was included in supplementary figure 3a,b to show the overall expression levels. The overall expression level of each LRRK2 proteins was quantified in supplementary figure 3b. In addition, immunofluorescence images of LRRK2 and the quantification of the expression levels were attached in supplementary figure 3c, d. The image shows that the overall distribution of LRRK2-positive cells is indistinguishable

among the experimental groups. These results are described in page 8 of the text.

OK

7) why is there no regular sized synuclein in the blot? Even in insoluble fractions, should be... unless there is none in the cell line? S can the authors also provide soluble? Or are cells internalising aggregated synuclein? It is also unclear from the text what synuclein is "relative" to - presumably actin control... would be better assessed relative to total synuclein in the cell?

Response: The soluble fraction blot has been attached in the revised manuscript. Soluble fraction blot showed the normal size a-synuclein. After treatment of a-synuclein-containing media, the aggregated forms of a-synuclein accumulated only in the triton-insoluble fractions. In soluble fraction, only LRRK2 G2019S-transduced cells showed the increase in the levels of monomeric a-synuclein. These are described in page 9. The levels of a-synuclein were normalized with those of actin, thereafter the relative levels of a-synuclein to the maximum levels of the internalized total a-synuclein in mock-transduced SH-SY5Y cells were calculated. The explanation on the quantification was included in the revised manuscript in methods (page 29).

OK, but there is still the need to explain why there is no monomeric synuclein in the insoluble fraction? See Volpicelli-Daley 2011.

Could it be that the triton insoluble fraction here is detecting only very high molecular weight aggregates (>100kd) that were only in the media, not the cells? A western blot of the secreted a-syn media alone may prove useful; I'd be very surprised and concerned if there were no soluble synuclein asides very high molecular weight aggregates?

Response: The most important differences between Volpicelli-Daley 2011 and our present study are the source of a-synuclein protein. The former study used in vitro fibrils generated from bacterially expressed protein, whereas our study used a-synuclein preparation secreted from neuroblastoma cells. These two preparations have both monomers and multimers (Supplementary Fig. 5), however, have different characteristics. When taken up by cells, both preparations also show both monomers and multimers. The in vitro fibrils contain monomers both in Triton-soluble and -insoluble fractions [Our own in vitro preparations also show the same property (see the figure below)], whereas the monomers of cell-secreted a-synuclein mostly go to the Triton-soluble (figure 2a), very little goes to the insoluble fractions (figure 2c) (This has been described in the revised manuscript, page 10). These results indicate that the cell-secreted a-synuclein multimers are more SDS-resistant than the in vitro fibrils. However, the exact nature of these differences is unknown.

8) Problems with data presentation: number of cells expressing seems inappropriate... all are 1 yet with variability? Most GS must contain 1.4 with 10% variance with odd cell being way off? Also, n is not 3 surely (as stated)?, n is likely 200 from 3 independent experiments. This needs to be much clearer throughout; change the writing or the presentation (scatter plots with overlaid mean +/- SEM would help)

Response: Yes, we realize that the data in Figure 2e are very confusing in its current form. The reviewer correctly understood the data that three independent experiment was performed; each experiment with n = 200. In the revised manuscript, graphs in Figure 2d and Figure 2e (in the original manuscript) were combined in Figure 2f with mean/SEM. The Y axis of figure 2f represents the number of BiFC (+) cells out of 200 cells, and the colors represent the number of BiFC fluorescent puncta in BiFC (+) cells. The explanation has been included in the figure legend (page 48).

Throughout this seems unclear, if 200 cells or worms (technical replicates) were compressed to a single datum for each independent replicate (x3) on would expect greater variation? Scatter plots instead of bar charts would help understand variance or not in the data and how statistics were performed (likewise F values reported with degrees of freedom and residuals)

Response: We agree with the reviewer that scatter plots would help better appreciate variance. In the revised manuscript, we have included the scatter plots of the data from each experiment in Supplementary Fig. 6. We decided to have the bar graph in the main Fig. 2f, because it is difficult to visualize the number of cell with scatter plots (there are too many

data points: 200 dots per each column). Another correction: number of experiments was actually 5 instead of 3. This has been corrected in the figure legend. F values with degrees of freedom were included in figure legends (page 52).

9) *No explanation of blue signal DAPI? Again to interpret such data we need to know TF efficiency of each, how many cells, how close they are to each other.*

Response: Blue signals indicate the nuclei, which is labeled with Topro3 iodide. The explanation has been included in the figure legend in revised manuscript (page 47, 48, 49). The cells we used in this study stably express a-synuclein-Venus N or C fusion proteins, so a half the cells express a-synuclein-VenusN, the other half a-synuclein-VenusC in co-culture of these cells (Bae et al., 2014, Nature Commun).

OK

10) *Need to address why inhibitor only works in OE not mock. Need to check levels of OE, density in the cells, proximity etc? Residual effects not altered by inhibitor... no endogenous LRRK2? ALSO mutant not different from BAC GS - this is not as expected if a kinase dependent phenomenon, expect expression levels may be different.*

Response: The levels of the endogenous LRRK2 is very low in SH-SY5Y cells; we were not able to detect the endogenous LRRK2 by western blotting. This is probably why it is difficult to see the effects of the inhibitor (page 10). The levels of transduced LRRK2 was quantified and attached in the supplementary figure 3. As you will see in this figure, the overall expression levels tend to be variable, however the differences are not statistically significant. The intracellular distribution of LRRK2 proteins were indistinguishable among the experimental groups. Note that we used adenoviral vectors, so most of SV2 cells were infected with the virus, which represent about 50% of the total cells in the co-culture. This is described in page 8. LRRK2 mutation D1994A is an artificial mutation which harbors the catalytically inactive kinase domain. Overexpression of LRRK2 D1994A was used as a negative control for the responses to the kinase inhibitor. Increased propagation of asynuclein by the overexpression of either LRRK2 WT or LRRK2 G2019S were reversed by the inhibitor treatment, whereas the propagation in the LRRK2 D1994A-transduced cells was not responsive to the inhibitor. These data confirmed that LRRK2 could enhance the propagation a-synuclein in a kinase activity-dependent manner.

OK

11) *Fig 3. Colocalisation experiments must be by coefficients (eg pearsons R) puncta analyses and overlap / coloc number are fine, but require extra information of puncta number, size and intensity to make sense of number colocalised. Similarly n is not clerly reported again (see point 8)*

Response: As per the reviewer's suggestion, pearson's coefficient was calculated and attached in revised manuscript in Figure 3c (page 11).

OK

12) Does *rab35* LRRK2 interaction occurs with non-overexpressed endogenous proteins also? Seems example blots are missing in Fig3 D? Also a casual glance makes it look like MORE *rab35* associated with LRRK2 in WT than GS in c & d (unclear in 3F10 example). This should be quantified as might be driving force for alterations, or at very least addressed in text

Response: We were not able to detect the endogenous LRRK2 in differentiated SH-SY5Y cells, and thus, we could not confirm the interaction between the endogenous LRRK2 and Rab35. Should we try IP in brain extracts?? The Quantified graph of the IP experiment has been attached in the revised manuscript (Figure 3e, g). There is no difference between the WT and GS mutant. However, since IP experiment is only semi-quantitative, we are hesitant to draw any serious conclusion over this result, rather we merely confirm the interaction between these proteins.

OK

13) other than n's being hard to decipher... statistical analyses inappropriate; presented for 2way RM ANOVA yet paired ttest???

Response: As per the reviewer's suggestion, the p values were calculated by one-way ANOVA with tukey's post hoc test, one-way ANOVA with Dunnet's post hoc test, or two way ANOVA with Dunnet's post hoc test throughout the revised manuscript when appropriate.

Incorrect. As stated above, graphs presented as 2way analysed by 1 way. Also one should not report two types of post test. Also may not be necessary if interactions are proven significant by 2way ANOVA e.g., genotype x treatment in fig 7.

Response: In revised manuscript, graphs in Figure 7 were analyzed by two way ANOVA with Tukey's post hoc test. P values and F values were included in figure legends (page 57-58).

14) Figure 4 n again hard to work out. Life span here and other two in WT shows they seem to mostly ALL die on the same day - here is day 16, p day 14, fig5h is 14. Other than Fig 4h is hard to know how much a difference (1day) is due to variability or treatments... scatters and join bars between replicates would help, e.g., in 3 experiments is WT always the same sine different? Is randomness in the 200?

Response: The Mean life span is very reproducible in *C. elegans*. WT worms (N2) have the mean life span of about 16 days. When V1S and SV2 were injected, the mean life span was reduced to about 13-14 days. In Figure 4h and 5h, the control columns represent V1S+SV2 lines in N2 and Rab35 backgrounds, respectively. To clarify this point, we added the V1S+SV2 labels in the figures. Survival curves from all individual experiments are shown in Supplementary Figures 1, 5, and 6 in revised manuscript, so the readers can appreciate the variability of this assay.

Scatter plots with replicate joins would be much more clear

Response: In addition to the survival curves with dots in Supplementary figures 2, 8 and 9, we have replaced the mean life span graphs with scatter plots in Figs.1h , 4g, 5g, and 6g.

15) Figure 6 labelled "transmission". How is this clearly transmission, and not reduced expression in certain cell types? Which cells are expressing, are they assaying in areas which don't express on the transgene?

Response: This experimental model, as explained in Figure 1a, expresses a-synuclein-VenusN and a-synuclein-VenusC in pharyngeal muscle cells and the connected neurons, respectively, using the cell type-specific promoters. We chose the transgenic lines based on the expression levels of a-synuclein, so that all the transgenic lines express similar levels of a-synuclein. Expression levels of a-synuclein in lines are shown in Supplementary Figures 1, 5, and 6 in the revised manuscript. Expression levels of Rab35 proteins are shown in Supplementary Figure 6d in the revised manuscript. The point about the expression levels is described in the text (pages 6, 11, and 12).

OK

16) Figure 6 Puncta analyses... again if percent coloc. need details on number and intensity of both signals. Better with Pearson's. That said rab increases in intensity are interesting so need to show more e.g., puncta size, number etc. rab levels by wb etc.

Response: In Figure 7 in revised manuscript, we have expanded the description of this figure in the Results section and provided additional and more analysis of the levels and size of the Rab35 positive structures (page 13-15). Moreover we have completely re-written this section to provide more clear description of the analysis and comparisons among groups for the various panels.

OK

17) Figure 6. What is the evidence for target engagement in these experiments (not others) e.g., LRRK2 phosphorylation. Also again the lack of detail on statistical analyses are alarming, was a stats form filled in? Where are details of the ANOVA F-values and df, residuals etc throughout?

Response: Indeed we considered the levels of LRRK2 phosphorylation represent an indirect measure of target engagement. We apologize for the lack of details in the statistical analysis. We have completely revise the Methods, Results and Figure legend sections as applicable. We performed one-way ANOVA with post-hoc Dunnet when comparing to non-tg vehicle and with Tukey-Kramer when comparing to the tg treated mice. Moreover we have now provided the detailed F and P values for each of the experiments.

OK but not fully. I'd also request scatter plots as they are much more informative of technical v independent replication. This is the strongest part of the manuscript, but again a 2way

ANOVA should have been employed

Response: In revised manuscript, graphs in Fig. 7 and Supplementary Fig. 10 were changed to scatter plots. The statistics were performed with two way ANOVA.

18) Supplemental fig 6. The inhibitor reduces LRRK2 p but also seemingly Rab35 and aSyn levels. While they show the rescue attempt is superseded by Rab constitutively active, these conclusions require evidence for whether LRRK2 protection is mimic by reduction in pRab?

Response: We also find it interesting that the inhibitors cause the reduction in levels of Rab35 and a-synuclein. Reduction in a-synuclein levels is consistent with the IHC data showing the reduction of the total a-synuclein deposition. This seems to be the result of the rescue effect of the kinase inhibitor. As much as we would like to measure the levels of phospho-Rab35 in the mouse brains, we unfortunately do not have tools to accomplish this at present. Although we tried IP of Rab35 and mass analysis to quantify the phospho-forms, we were not able to detect phospho-Rab35. It is well-known that phosphoproteins are difficult to ionize. It is a technical problem we should overcome. However, it will take us a significant amount of time and effort to solve this problem, which will significantly delay the publication of the work. In the revised manuscript, we stated that measurement of phospho-Rab35 has not been successful in mouse brains and needs to be addressed (page 14).

As for the reduction in Rab35 levels, it could be due to either increased degradation or reduced synthesis of Rab35 protein. The role of LRRK2-mediated Rab35 phosphorylation in the reduction of its levels would be another interesting topic for investigation. As much as it is interesting and important, this issue is better addressed in a separate study in the future.

OK – but elaborated discussion of these facts would help.

Response: Discussion on these issues have been included in the revised manuscript in pages 23.

Reviewer #3 (Remarks to the Author):

Below are comments from Reviewer #1 and Reviewer #3 pertaining to the original manuscript.

Reviewer #1

The term 'intercellular transmission', for example in the title and in the text, is verbose compared to the term commonly used in the field- 'propagation'. Further, the word

'transmission' may not be very accurate, because the aggregates themselves may not 'transmit' as it may be assumed in this work (yet not demonstrated). The authors and many others can clearly show a-synuclein inclusion propagation through the brain in rats and mice and in vitro. It is advised to revise the relevant portions of text.

Au Response: As per reviewer's suggestion, the term 'intercellular transmission' has been changed to 'propagation' throughout the text. The title is now 'LRRK2 kinase regulates propagation of a-synuclein via phosphorylation of rab35.'

Reviewer #3

Between lines 83-102 there is no data that shows that a-synuclein aggregates were transmitted. How was it ruled out that α -synuclein monomers exchanged between the nerve and muscle cells in wild type worms? Perhaps in the lrk-1 worms the mutation inhibits release of the monomer. More on this point, in Figure 1a, the right-hand images of the donor and recipient cells shows that both cells release monomers, which are not aggregates.

Response: Based on the cells studies (e.g., Lee et al., 2005, J. Neurosci; Jang et al., 2010, JNeurochem), both monomers and multimers (dimer/trimer/oligomers) are secreted from cells. Although the templated seeding mechanism postulates that the high-order multimers are the ones to serve as seeds for aggregate transmission, it has not been properly demonstrated.

Revised manuscript:

Reviewers 1 and 3 brought up in essence the same issue, i.e., "the aggregates themselves may not transmit" and "how was it ruled out that a-syn monomers [and not aggregates] exchange between nerve and muscle cells?" But the authors did not adequately address this issue.

The worm and SY-SH5Y data can be explained by V1S-a-syn monomers being secreted by one group of cells in the co-culture and then being taken up by cells expressing a-syn-SV2 monomers, or vice versa. Additionally, one cell type could release V1S-a-syn monomers and the other cells in the co-culture could release a-syn-SV2 monomers, resulting in the generation of Venus fluorescence in the media from the V1S-a-syn/a-syn-SV2 dimer. The data in Figure 2 especially, and the related text, would greatly benefit from the authors explicitly stating that their data can also be explained by monomer propagation between cells. The authors have not proven in their revised manuscript that a-syn aggregates are the main species that transmit/propagate from one cell to another.

Response: The reviewer's points are well taken. We agree on the issue that one cannot say for sure which species are transmitted. In the revised manuscript, this issue has been discussed in pages 21. As for the possibility of monomers forming dimer in the media, we ruled it out in an experiment where culture media from V1S and SV2 cells were separately collected and mixed. This experiment did not result in Venus fluorescence. This result was added as Supplementary figure 11 in the revised manuscript.

REVIEWERS' COMMENTS:

Reviewer #2 (Remarks to the Author):

The authors have done a competent revision and should be commended, although it is a shame it took 3 rounds.

Scientifically, and in terms of interpretation, the document is now much clearer and will add significantly to the field.

I disagree with the suggestion that the document has been proof read well for English errors, but feel it is now the task of the editorial staff to address this, not peer review. To aid this I include a marked pdf of the document, but this is not comprehensive as I don't think it should be a task for reviewers.

Reviewer #3 (Remarks to the Author):

You have addressed my concerns. This version is much better than the original.

REVIEWERS' COMMENTS:

Reviewer #2 (Remarks to the Author):

The authors have done a competent revision and should be commended, although it is a shame it took 3 rounds.

Scientifically, and in terms of interpretation, the document is now much clearer and will add significantly to the field.

I disagree with the suggestion that the document has been proof read well for English errors, but feel it is now the task of the editorial staff to address this, not peer review. To aid this I include a marked pdf of the document, but this is not comprehensive as I don't think it should be a task for reviewers.

Response: Our manuscript has been edited by a professional editing service, Nature Research Editing Service.

Reviewer #3 (Remarks to the Author):

You have addressed my concerns. This version is much better than the original.